# Separating CO$_2$ emission from removal targets comes with limited cost impacts

Anne Merfort [1,2] ✉, Jessica Strefler[1], Gabriel Abrahão [1], Nico Bauer [1], Tabea Dorndorf[1,3], Elmar Kriegler[1,4], Gunnar Luderer [1,2], Leon Merfort [1,2] & Ottmar Edenhofer [1,5]

Net-zero commitments have become the focal point for countries to communicate long-term climate targets. However, to this point it is not clear to what extent conventional emissions reductions and carbon dioxide removal (CDR) will contribute to net-zero. An integrated market for emissions and removals with a uniform carbon price delivers the economically efficient contribution of CDR to net-zero. Yet it might not fully internalise sustainability risks of CDR and hence could lead to its overuse. In this study, we explore the implications of separating targets for emissions and for removals delivered by novel CDR in global net-zero emissions pathways with the Integrated Assessment Model REMIND. We find that overall efficiency losses induced by such separation are moderate. Furthermore, limiting the CDR target comes with increasing emission prices but also significant benefits: lower cumulative emissions, a lower financial burden for public finance of CDR and limited reliance on geologic CO$_2$ storage but fails to lower the biomass demand. Proposed targets should also ensure sufficient CDR deployment to achieve net-negative emissions in the second half of the 21st century.

Net-zero emissions pledges have become a central means to communicate long-term emission reduction commitments in international climate policy[1]. As of April 2024, 148 countries causing 88% of global GHG emissions communicated a net-zero emission target[2] motivated by the conclusion of the IPCC's Special Report on 1.5 °C that global net-zero CO$_2$ emissions have to be achieved in the early 2050s to limit global mean temperature increase to 1.5 °C by 2100 with low overshoot[3]. To reach a net-zero CO$_2$ target, Carbon Dioxide Removal (CDR) will be necessary to compensate all residual CO$_2$ emissions, i.e., the amount of gross CO$_2$ emissions from fossil-fuel combustion, industry processes and land-use change (before CDR is deployed), of which abatement remains uneconomical at the CO$_2$ price corresponding to a given reduction target.

While CDR methods play a significant role in climate change mitigation pathways, as of today both industrial scale-up as well as international and national legal frameworks and policies lag behind the envisioned CDR deployment in 1.5 °C scenarios[4]. Furthermore, especially so-called novel CDR (nCDR) methods, which could store CO$_2$ out of the atmosphere for centuries to millennia with low risk of reversibility are still at low technological readiness levels and not yet proven at large scale[4]. How much CDR will be feasible, and how scale-up should be incentivized vis-à-vis emission reductions are key questions of recent debates.

In the literature, a variety of policy instruments and commercialisation options are discussed to incentivise CDR, which are categorised by Hickey et al.[5] into market-based approaches, public procurement schemes and fiscal incentives and include mechanisms such as carbon pricing, carbon markets or tradable obligation schemes, results-based payments and subsidies or tax credits.

[1]Potsdam Institute for Climate Impact Research (PIK), Member of the Leibniz Association, Potsdam, Germany. [2]Global Energy Systems Analysis, Technische Universität Berlin, Berlin, Germany. [3]Geographical Institute, Humboldt Universität zu Berlin, Berlin, Germany. [4]Faculty of Economics and Social Sciences, University of Potsdam, Potsdam, Germany. [5]Climate Economics and Public Policy, Technische Universität Berlin, Berlin, Germany. ✉e-mail: anne.merfort@pik-potsdam.de

From a purely economic perspective, a uniform carbon price in all sectors and on all emissions is, in absence of other externalities, the economically efficient solution[6], which would be delivered by an integrated market for emissions and removals. For incentivising CDR this means: $CO_2$ emissions should be avoided to the point, until it is cheaper to compensate the next ton of $CO_2$ by CDR. That is, the price paid for CDR should be equal to the carbon price on emissions, such that marginal abatement costs equal the marginal supply costs of CDR.

This, however, only holds true if $CO_2$ removals and emission reductions are regarded equivalent in their role for net-zero, which is problematic for a variety of reasons, especially when removals are to be delivered by the land-use sector[7]. CDR methods are associated with environmental side effects[8] that pose risks for sustainability[8,9], such as potential leakage, toxicity of chemical sorbents or mineralisation methods or land-impacts on biodiversity, water availability or food security from biomass demand. Intertemporal equivalence, i.e., allowing continued emissions now to be offset by future removals, is seen especially problematic[7]. It could provoke mitigation deterrence, if expectations of future mitigation through nCDR justify reduced near term emission reduction efforts, which could ultimately lead to underachievement of climate targets if CDR methods fail to deliver[10]. By challenging the equivalence of emission reductions and removals, multiple scholars advocate for a separation of targets[7,11–14].

If targets on emissions and removals were to be set by the regulator, in contrast to the case where the quantities emerge as outcome of an integrated market with uniform prices, deviations from equal prices will likely arise; either explicitly in a carbon pricing policy setup or as 'effective' carbon prices of other policy options. The main concern with spelling out separate targets is the entailed deviation from the market efficient solution and the associated efficiency losses.

However, price equalisation is only cost-efficient in the absence of market externalities and in case of non-strategic actors with perfect foresight. Yet, carbon markets may not adequately price sustainability risks of CDR[8,9], creating a misalignment between market outcome and societal objectives. Hence, a socially optimal contribution of CDR might be lower than the efficient outcome and a separate target for CDR might be a tool to achieve a socially optimal contribution.

Furthermore, the lack of explicit targets leaves expectations of future CDR acceptability, desirability and ultimately availability unclear. This lack of clarity could reduce planning security for fossil emitters risking a lack of security for zero-carbon investments. This could provoke strategic behaviour and lobbying; in the near-term if fossil fuel emitters oppose the necessary speed of the transition and also in the long-term if expectations about large amounts of future removal capacities were frustrated and carbon prices jump sharply. At the same time, the lack of clarity reduces planning security for CDR investors, yet ambitious CDR commitments are needed now to enable net-negative emissions in the second half of the century. Therefore, separate targets are proposed to manage expectations as a means to prevent mitigation deterrence[15], ensure sufficient CDR development to achieve net-negative emissions later and enhance political credibility of net-zero targets[12,13], which might be more important for effective climate policy than economic efficiency[16].

A third perspective has so far received less attention within the debate about separate targets on emissions and removals. If a uniform carbon price is used to remunerate removals and penalize emissions, this could lead to huge windfall-profits[17] especially if CDR-specific deployment constraints or market externalities or imperfections are present, such as environmental side-effects and technological learning impacts. In general, these windfall-profits could be taxed away with well-designed rent taxation. However, if rent taxation is politically not feasible, differentiation of carbon prices might be justified[18] and could therefore support a separation of targets.

So far, studies found that a removal price below the price on emissions is optimal, if $CO_2$ is not stored permanently[18,19]. On the other hand, Franks et al. found that a lower risk for interregional leakage for some CDR methods in comparison to $CO_2$ abatement could render a removal price greater than the emission price optimal[20].

It should be noted that while a separation of targets may lead to price differentiation, in reverse a price differentiation between emissions and removals does not exclude the integration of CDR in an Emission Trading Scheme as outlined in a recent report by the European Scientific Advisory Board on Climate Change 2025[21]. Furthermore, Rickels et al.[22] propose removal certificate reserves to ensure price stabilisation while Lessmann et al.[23] propose so-called clean-up certificates for future removals of current emissions for integration into the European Emissions Trading Scheme.

Although the role of residual emissions and CDR in net-zero pathways has been discussed[24–26], adopting explicitly specified targets for the two components into net-zero pathways provides a useful tool for assessing the implications of this policy framework.

In this study we therefore integrate separate targets for emission reduction and removals into an Integrated Assessment Model (IAM). While an IAM by itself is not able to answer the question whether targets for emissions and removals should be separated, we investigate the trade-off with economic efficiency if targets were separated. For that we analyse the consequences of adding a separate target for nCDR, that may cause deviations from the market efficient contribution of nCDR to net-zero. We address consequences on mitigation pathways as well as economic consequences of such a policy, which is not possible in a conventional set-up with an integrated carbon market and a uniform carbon price but constraints on CDR. From our analysis we derive policy recommendations on how to set separate targets in the face of uncertain future developments using the IAM REMIND[27].

## Results

### Net-zero quantity goals and separate CDR targets

Using the IAM REMIND[27,28] with a detailed representation of the global energy system we design different climate change mitigation scenarios that achieve global net-zero $CO_2$ in 2050. We explicitly prescribe varying quantity targets for residual $CO_2$ emissions (i.e. all $CO_2$ emissions from fossil fuel combustion, industrial processes and land-use before nCDR) at the time of net-zero that have to be compensated by the corresponding amount of nCDR. The model's available nCDR options are Direct Air Carbon Capture and Storage (DACCS), Bioenergy with Carbon Capture and Storage (BECCS), Enhanced Weathering of rocks (EW) and Industry CDR (Industry BECCS or carbon capture and storage from fossil-free synthetic fuels) (see Methods for more details on emissions and removal accounting and Supplementary Methods as well as Supplementary Tables 1-4 for underlying technology data).

Throughout this modelling exercise we presume that targets for emission reduction and nCDR were to be set separately without the possibility of later adjustment. As the market efficient outcome is difficult to predict with current knowledge, policymakers may choose targets that diverge from what would emerge from an integrated market with equal prices. Therefore, we span the scenario range from 2 to 12 Gigaton $CO_2$ per year (GtCO$_2$ per yr) of residual emissions (corresponding to -5% up to -27% with respect to 2019 global $CO_2$ emissions[29], see Supplementary Note 1 for the underlying rationale) and the same amount of compensating nCDR in 2050.

The scenario with 7 GtCO$_2$ per yr as quantity targets for residual emissions and nCDR is the scenario from which identical prices emerge; i.e. the regulator perfectly guessed the market efficient contribution of nCDR to net-zero. Note that in our model this is identical to a scenario with an integrated market and a uniform carbon price due to the model features, namely perfect foresight, certainty of future costs and a social planner. This 7 GtCO$_2$ per yr scenario serves as a benchmark throughout the manuscript.

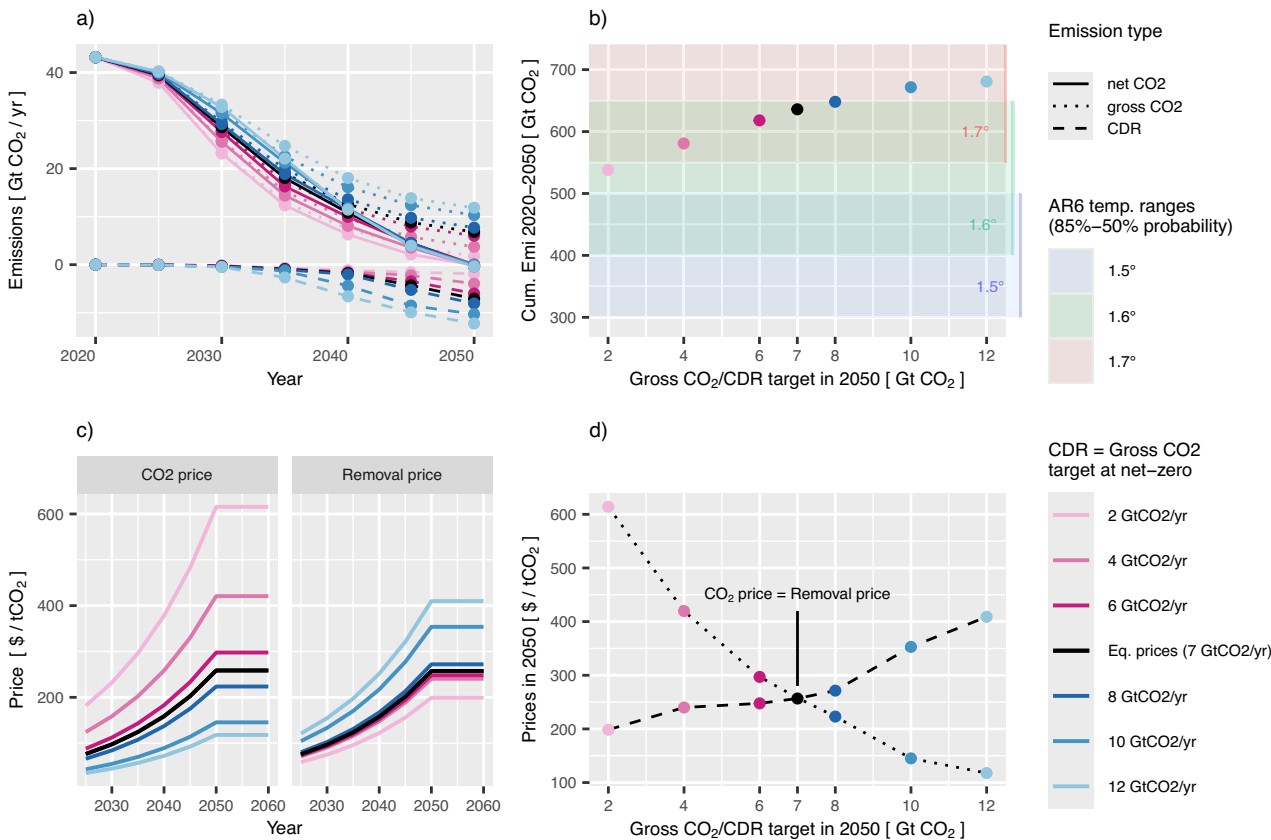

**Fig. 1 | Overview of CO₂ emissions and CO₂ prices in scenarios with differentiated regulation of residual CO₂ emissions and novel Carbon Dioxide Removal (nCDR).** Global net-zero CO₂ targets with varying amounts of residual emissions and nCDR lead to differences in the emission pathways until net-zero and diverging prices. Stronger targets on residual emissions lead to earlier decarbonisation (**a**) and lower cumulative emissions (**b**) under Hotelling price path assumptions (**c**). Furthermore, the price on emissions is more sensitive to the reduction target than the removal price is on the respective nCDR target (**d**). All monetary values are in 2005 US dollars. Source data are provided as a Source Data file.

Both the endogenously derived shadow prices on emissions (hereafter short: carbon price) and for nCDR (hereafter short: nCDR price) follow a Hotelling price path with a growth rate of 5% per year until the time of net-zero in 2050 and remain constant thereafter (see Fig. 1c). The carbon price is also applied to non-CO₂ greenhouse gases, leading to substantial but across scenarios almost identical non-CO₂ GHG emission reductions that will not be further discussed here (see Methods).

Re- and afforestation are prominent CDR methods in mitigation scenarios and also available to REMIND, but for conceptual clarity we exclude them from the quantity target. Separating prices for de- and re/afforestation must be treated with special care as it can lead to perverse incentives for unsustainable management. A clear example is a removal price payed for afforestation that is higher than the price on emissions caused by deforestation, which would incentivise clearing of existing forests for reforestation.

In our analysis, re-/ afforestation follows exogenous assumptions that are identical across scenarios and the net effect of total land-use change emissions is fully accounted for in the residual emissions.

**Emission pathways to net-zero and carbon prices for separate targets**

First we discuss the variations of emission trajectories and corresponding carbon and removal prices between the scenarios with different net-zero formulations. Gross CO₂ emissions diverge already in 2030 due to different carbon prices reflecting the decarbonisation ambition in 2050. Yet nCDR scale-up takes time, primarily due to the

high upscaling rates needed from close to zero nCDR deployment to date[4] and the need for significant future cost reductions due to technological learning. Hence, climate-relevant amounts are only reached in 2040 and beyond (see Fig. 1 a). The different dynamics of emission reduction and the scale-up of CDR deployment lead to different cumulative emissions (Fig. 1b). In fact, the cumulative CO₂ emissions from 2020 to 2050 range from 538 GtCO₂ in the scenario with 2 GtCO₂ per yr (residual emissions and nCDR in 2050) up to 680 GtCO₂ in the scenario with high reliance on CDR (12 GtCO₂ per yr), even though net CO₂ emissions reach zero at the same time.

The carbon price on emissions varies strongly depending on the level of residual emissions across the full scenario scope (Fig. 1 c). We observe more than a 5-fold increase from the scenario with largest reliance on nCDR (12 GtCO₂) with 120 US\$(2005) per ton CO₂ (hereafter \$ per tCO₂) to the scenario with little nCDR deployment and the most ambitious reduction (2 GtCO₂) with 610\$ per tCO₂. This is in line with Knopf et al.[30] that find non-linearly increasing challenges to mitigation with increasing mitigation efforts. On the other hand, the carbon price for nCDR is less sensitive and only doubles across the full scope of scenarios, ranging from 200\$ per tCO₂ under little reliance on nCDR (2 GtCO₂) to 410\$ per t CO₂ in the scenario with strong nCDR deployment (12 GtCO₂) (see Fig. 1 d). Furthermore, most of the price increase only occurs for quantity targets beyond 8 GtCO₂ when DACCS enters the CDR portfolio while for the range of lower quantity targets the price on nCDR remains remarkably flat.

The main reason for the lower price sensitivity of CDR is that even for a low CDR demand scenario there are no low-cost nCDR options

available. This is, because sustainable biomass is always limited, and the high demand for biofuels in low CDR demand scenarios forces the more expensive Fischer-Tropsch-BECCS technology into the CDR portfolio. On the other end of the spectrum, DACCS is an expensive but scalable option, and higher demands do not increase prices as much, due to two reasons. DACCS deployment per unit becomes cheaper with larger quantities due to learning and, by scenario design, a higher DACCS demand is paired with a less stringent target on emission reduction, relaxing competition for renewable energy, a key resource for large-scale DACCS deployment.

## nCDR deployment and impacts of different net-zero formulations

We observe only very small contributions of nCDR in 2030 across scenarios, as scale-up and technological advancement take time. However, having large amounts of nCDR (>6 $GtCO_2$ per yr) available in 2050 requires earlier scale-up, which translates to higher quanitites of up to 500 $MtCO_2$ per yr CDR already in 2030. Since almost all of this requires CCS, achieving such high amounts already in 2030 would require an immediate and global effort. For example, in the Net-Zero Industry Act[31], the European Commission has proposed that the EU develops at least 50 million tonnes per year of $CO_2$ storage capacity by 2030, which primarily aims to cover industrial process emissions and will likely not be available for CDR. This CCS target is already ambitious, especially compared to the total capacity of projects within the EU that are operating or under construction as of 2023 of only 2.65 $MtCO2$ per yr[32] which requires annual growth rates of 52%. However, this target is only a tenth of what might be needed for CDR alone underlining the risks associated with a too high reliance on future CDR availability.

Removal prices in the 10 and 12 $GtCO_2$ per yr scenarios in 2040 are already high enough (>200$ per $tCO_2$) to incentivise significant contributions from EW. This is due to the fact that EW deployment relies on infrastructure for mining, grinding and transportation of material that already exists today and therefore EW could be scaled up in shorter time periods, yet regulatory and legal frameworks are largely missing. BECCS and EW are the major contributors to fulfilling the CDR targets in 2050, contributing similar shares except for the 2 $GtCO_2$ per yr (mostly BECCS). This potential early contribution of EW to permanent carbon removal suggests that EW should receive more attention as a component in regional CDR portfolios. Increasing the CDR quantity target, we find increasingly larger contributions from BECCS options with higher capture efficiency, such as $H_2$ and electricity production in addition to bioliquids.

The deployment of specific technologies and their relative contribution can depend strongly on the CDR target. EW is deployed in all scenarios except the one with the lowest CDR quantity target, and 2050 deployment scales up almost linearly with increasing CDR target. Industry CDR has in all scenarios a similar, but small contribution to overall removals. DACCS is only deployed in scenarios with quantity targets above 8 $GtCO_2$ per yr and is accompanied by a significant increase in the necessary CDR price due to its high costs. Note that we focus on global targets, and that at a regional level DACCS may be needed to reach country-level net-zero even for very low CDR targets. Furthermore, there are constraints to deployment that are not captured by the model such as the availability of critical raw material, public opposition or the lack of regulatory frameworks, that could severly constrain future CDR deployment.

Total biomass use is lowest in the equal-pricing net-zero formulation and increases stronger for high nCDR targets (blue scenarios) due to higher BECCS demand. But also for low nCDR targets (pink scenarios) (Fig. 2b) biomass demand increases due to increased pressure to decarbonise remaining liquid fuels that can no longer be offset by CDR. Note that total biomass use in 2050 is already close to the exogenously imposed sustainability limit of 100 Exajoule per year (EJ

per yr) across the whole scenario range and all scenarios exploit the full potential shortly after net-zero and throughout the second half of the century.

We find a quasi-linear relation between the 2050 gross $CO_2$ reduction target and the remaining fossil primary energy of approximately 15 EJ per yr increased fossil fuel use per $GtCO_2$ per yr residual emissions at net-zero, corresponding to a reduction of 60–93% from fossil fuel use in 2020. Further information on the level of residual emissions and especially their distribution across sectors can be found in Supplementary fig. 1.

Although available to the model, we do not observe fossil carbon capture in any of the scenarios, due to substantial residual emissions from imperfect capture and upstream $CH_4$ emissions[33] and the competition with nCDR for the carbon transport and storage infrastructure. We observe a quasi-linear increase in geologic carbon storage with a stronger increase for the highest CDR targets of 10–12 $GtCO_2$ per yr when DACCS becomes viable. Interestingly, the total volume of captured carbon, exhibits similar magnitudes of around 5–6 $GtCO_2$ per yr across nCDR targets between 2 and 8 $GtCO_2$ per yr. In low nCDR scenarios (2–4 $GtCO_2$ per yr), the amount of carbon captured that exceeds the CDR limit is not stored, but used to provide carbon-neutral synthetic fuels to substitute conventional liquids. Hence, even a low nCDR target cannot mitigate all risks associated with large-scale CDR deployment. While it could limit the dependency on geologic $CO_2$ storage, it does not relieve the pressure on biomass demand or carbon capture, as these are (under otherwise identical background assumptions) needed to decarbonise remaining liquid fuels.

Note that CDR reliance can be reduced significantly by demand-side mitigation options that essentially reduce total energy or food consumption[25]. Yet, this study focuses on the effect of target separation and we compare CDR contributions across scenarios with otherwise identical drivers. Hence, we do not include additional demand reductions other than what arises endogenously from the model as a cheaper option compared to alternative decarbonisation strategies (see Supplementary fig. 1).

## Fiscal and economic consequences of high and low nCDR contributions to net-zero and associated risks

Here we analyse the fiscal and economic consequences that arise from diverging prices in the emission and removal markets[18] and discuss the potential risks of setting targets 'inefficiently' with higher (blue scenarios) or lower (pink scenarios) contribution of nCDR to net-zero.

To quantify the efficiency losses we calculate consumption losses. But as scenarios differ in their cumulative emissions (Fig. 1), we cannot directly compare the consumption losses across the scenario range, because lower cumulative emissions generally result in higher consumption losses regardless of whether emission and removal targets are separated or not. To isolate the consumption losses introduced by the separation of targets (and the deviation from equal prices) from the consumption losses caused by achieving lower cumulative emissions (Fig. 1), the additional consumption loss is calculated with respect to counterfactual scenarios that are not shown here but described in the Methods. These scenarios achieve the same respective cumulative $CO_2$ budget until 2050 but with a uniform carbon price on emissions and removals (see Methods). We find that for moderate deviations (2–10 $GtCO_2$ per yr) from the equal-pricing contribution of nCDR to net-zero we observe only moderate efficiency losses of <10% additional consumption loss (Fig. 3). In absolute terms, it is an increase from 2.6% to 3.1% of consumption loss compared to a scenario with only current policies for the 12 $GtCO_2$ per yr (from 2020-2050 with respect to continued current policies) and only 4.3% to 4.5% in 2 $GtCO_2$ per yr scenario. This is in line with Strefler et al.[34] who also found only moderately increasing mitigation costs for moderate limitations on CDR in a uniform carbon pricing framework.

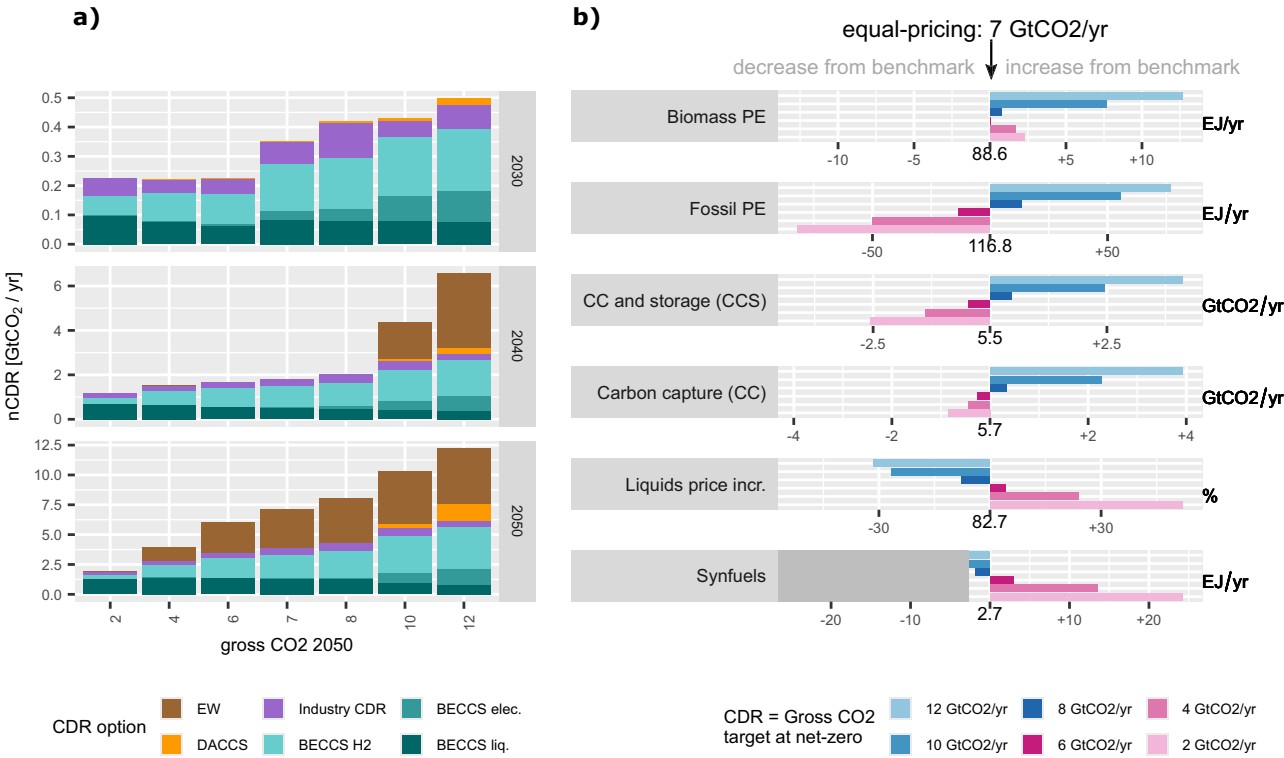

**Fig. 2 | Carbon Dioxide Removal (CDR) deployment portfolio and energy system indicators.** CDR deployment by technology in 2030, 2040 and 2050 respectively (panel **a**). Note the different scales of y-axes between panels. Panel **b** displays deviations of energy system indicators with respect to the scenario with economically optimal CDR (7 GtCO₂ per yr) (biomass and fossil primary energy (PE), carbon capture (CC) and carbon capture and storage (CCS), liquid fuels price increases in percent from 2020 to 2050 and total synthetic fuels (synfuels) demand. All

scenarios have a sustainability limit on total lignocellulosic biomass supply of 100 EJ per yr, which is fully exploited in all scenarios in the second half of the century. Note that CC refers to the total amount of captured carbon with yet undefined destination (CCU or CCS). CCS refers to the amount of captured carbon that is stored geologically, and CCU refers to the complement that is not stored but used to produce synthetic fuels. Source data are provided as a Source Data file.

If nCDR targets exceed the volume that would emerge in an equal-pricing case (blue scenarios), it will lead to a situation where the price on removals is larger than the price on emissions and therefore total nCDR expenditures exceed total annual emissions tax revenues, leading to a heavy burden on taxpayers (Fig. 3). The total carbon market value - the cumulative, discounted difference of CO₂ tax revenues and expenditure payed to remunerate removals with the removal price from 2020 to 2050 - would also be much smaller, as we find a decrease of carbon revenues and an increase of the nCDR expenditures with increasing nCDR targets. This strongly reduces the financial leeway for policy makers to support the transition, e.g., by subsidizing technologies or infrastructure or by redistributing revenues to ease regressive effects on poorer households.

The lower the target on residual emissions is (pink scenarios), the higher are the necessary near- and long-term emission prices and larger transitional challenges arise, which may lead to political pressure from high-emitting actors calling for a relaxation of the reduction target. Therefore, to avoid societal opposition and smooth out transitional challenges policymakers may understate the necessary reduction ambition and rely more on nCDR for achieving net-zero.

If non-market co-benefits of large-scale nCDR deployment outweigh the sustainability risk, a removal price above the emission price would be justified. In that case (blue scenarios) the necessary price on CO₂ emissions is lower, yet it entails crucial harms: higher residual emission targets lead to less near-term reductions that result in larger cumulative emissions (Fig. 1) and lower emission reductions have to be compensated by more nCDR, leading to the risk of missing the climate target entirely if nCDR does not deliver as expected (Fig. 3).

## Discussion

The idea of separate targets gained traction in the EU policy debate: The majority of stakeholders responding to the Public Consultation on the EU Climate Target for 2040[35] advocated for separate targets on GHG emission reduction, industrial removals (i.e. nCDR) and nature based removals.

There are two main lines of reasoning in the literature for separating targets for emissions and removals: Firstly, many scholars argue, a separation could prevent mitigation deterrence[10–12,16]. Because it could strengthen trust in political commitments[11,12], enable evaluations of ambitions in mitigation plans[36] and stabilise expectations. Separate targets could set a clear signal for the pace and depth of the phase-out of fossil fuels[12,13] and increase planning security for CDR suppliers to ensure sufficient CDR development such that net-negative emissions can be achieved after net-zero. Secondly, a separation of emission and removal targets is necessary, if damages from environmental side effects[8,37,38] are not reflected in the prices but are valued higher than the loss of economic efficiency from deviating from the economic optimum.

If the main concern is the avoidance of mitigation deterrence and the enhancement of policy credibility, then targets have to be binding and non-negotiable. On the other hand, if environmental side-effects are the main concern, a mechanism for iterative adjustments of the two targets, once more knowledge on side effects and future costs becomes available, might be an option to increase intertemporal flexibility.

In contrast, an integrated market with uniform prices is advocated for due to economic efficiency and the high uncertainty about future

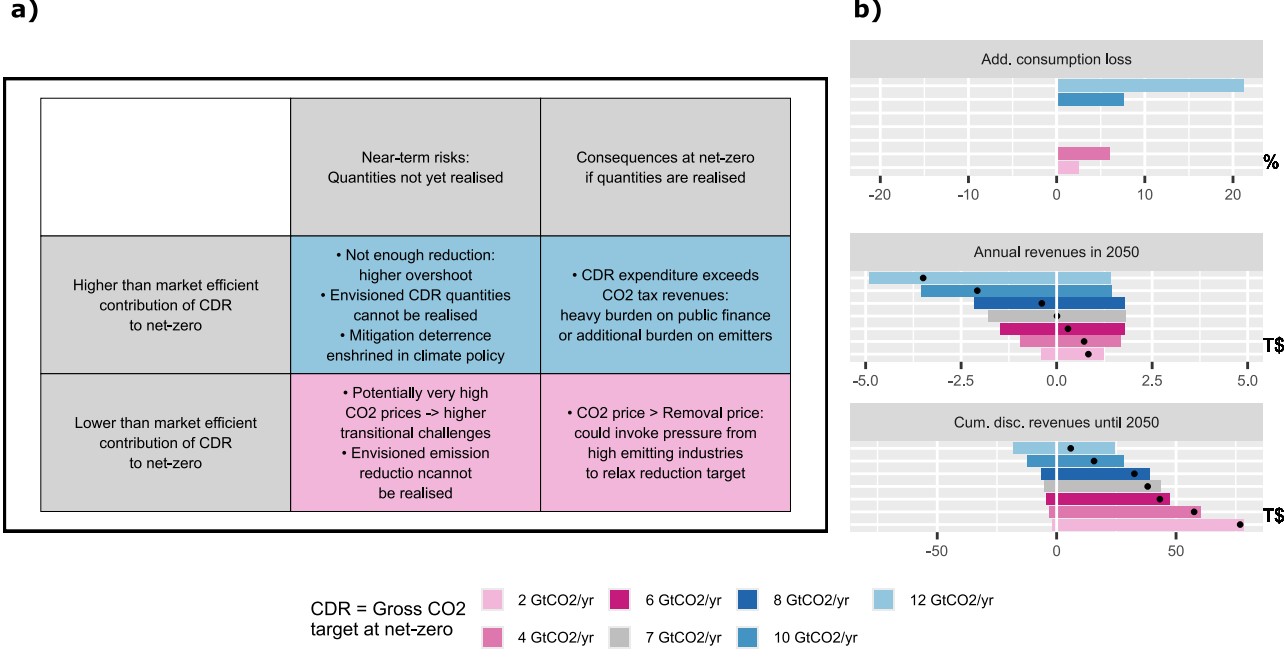

**Fig. 3 | Challenges that arise from higher or lower nCDR contributions to net-zero. a** Risks associated with higher (blue) or lower (pink) contributions of nCDR to net-zero with respect to the equal-pricing scenario. We find generally greater risks for high nCDR contributions. Note however, that these risks are not easily comparable and strongly depend on an individual risk perception. **b** Indicators for fiscal challenges: additional consumption loss with respect to counterfactual scenarios, reaching the same cumulative $CO_2$ budget until 2050 but with a uniform carbon price on emissions and removals (see Methods); annual net-revenues at the time of net-zero, i.e., total revenue from emissions pricing minus total expenditure on nCDR; carbon market value, i.e., cumulative (2020-2050), discounted (w.r.t. 2020 and at discount rate 5%) net-revenues. All monetary values are in 2005 US dollars. Source data are provided as a Source Data file.

quantities that makes target setting now difficult[39]. Additionally, scholars have started to outline how CDR could be included into already existing legal frameworks, such as the EU-ETS[23,39,40], as separate regulations are still underdeveloped[39]. Comprehensive and solid legal frameworks are a precondition for fast nCDR deployment, such as liability for storage integrity[41], MRV (monitoring, reporting, verification)[42] or carbon accounting of removals[43]. An important step foward is the EU Carbon Removals and Carbon Farming Certification (CRCF) Regulation (Regulation (EU) 2024/3012)[44] that was published in November 2024. This regulation establishes a voluntary certification framework for permanent carbon removals, carbon farming, and carbon storage in products within the EU by setting quality criteria, verification rules, certification scheme requirements, and certified unit issuance rules.

The REMIND modelling framework is not equipped to capture many of the externalities and market imperfections raised above; i.e. myopic investments, strategic actors and uncertainty about future costs and potentials of CDR methods. Therefore, we do not aim to answer the question whether targets should be separate. Instead we assess the consequences of setting targets and with that potentially diverging from the market efficient contribution of CDR to net-zero. We find that efficiency losses due to the separation of targets are moderate, suggesting that this prominent argument for equal prices may be overestimated.

We also provide insights how targets should be set if they were to be separate and highlight that economic efficiency losses and associated risks are larger for high-CDR scenarios. Namely, if targets were chosen with lower than equal-pricing CDR contributions and more stringent reduction targets, economic efficiency losses induced by a deviation remain limited at less than 10% additional mitigation costs. Furthermore, cumulative emissions remain lower, which reduces the

overshoot of the 1.5 °C limit. Due to higher revenues from carbon pricing and lower overall spendings on CDR there is more financial leeway for policy makers to mitigate regressive effects of climate policies on poorer households[17,45] or for green investments. We observe a steeper fossil fuel phase out that is accompanied by an increased reliance on biogenic and synthetic fuels to decarbonise the remaining liquids demand of the energy system. While these scenarios show a lower reliance on geologic storage of $CO_2$, we still observe a high demand for biomass due to increased pressure on the mitigation side.

A sensitivity analysis exploring unconstrained biomass use across net-zero target formulations reveals that biomass use is almost identical for a wide range of scenarios 2–8 $GtCO_2$ per yr. Note, however, that in low CDR scenarios a more stringent climate outcome is achieved with the same amount of biomass. When correcting for different climate outcomes, biomass use increases with increasing CDR target (see Supplementary fig. 2). Hence, the CDR target has an impact on overall biomass use but is by far not the sole driver of exacerbated biomass demand. A low nCDR target alone might not be enough to limit sustainability risks typically associated with large-scale CDR deployment particularly on land and additional land-use policies will therefore be needed[46].

Finally, the necessary emission price increases non-linearly with increasing reduction target strictness, posing aggravated transitional challenges, raising concerns if societies would continue to support mitigation policies. Ambitious emission reduction targets reduce reliance on CDR and therefore mitigate the risk of missing the climate target if large-scale CDR deployment should fail, but they also entail increasing risks of a failure of necessary emission reductions.

On the other hand, nCDR contributions to net-zero that are higher than the market efficient solution would only be socially optimal, if

there are non-market co-benefits that outweigh the non-market side-effects of large-scale nCDR deployment. So far, there is no such evidence in the literature. In that case, annual CDR expenditures at net-zero exceed revenues from emissions pricing, imposing a heavy burden on taxpayers and public funding. In case of very high CDR targets, mitigation costs increase significantly. The associated price on emissions is lower, easing transitional challenges but resulting in higher cumulative emissions and therefore higher climate impacts, leading to additional costs from climate damages.

Large uncertainties on future costs of emission reduction and CDR deployment exist. 1.5 °C mitigation pathways assessed by the 6th assessment report of the IPCC (AR6, WG3 Chapter 3)[47] show a large spread in carbon prices (see Supplementary fig. 3) as well as CDR contributions to net-zero. While we present relevant insights on the consequences of deviating from the market efficient solution, the absolute numbers are inherently uncertain. Even more so, global numbers presented in this study cannot be downscaled to regional levels. Special attention has to be paid to the fact that the emission accounting as well as the scope varies between this study and the net-zero targets adopted by many Annex I countries. Here we analyse global net-zero $CO_2$ but many countries include non-$CO_2$ emissions but also additional carbon flows from managed forests in their pledge to net-zero. In particular, significant discrepancies between country level land sink accounting and accounting in IAMs currently exist due to inconsistencies in the definition of anthropogenic flows from forests[48].

We call for the interdisciplinary scientific community to evaluate and negotiate between different risk perspectives. The political objectives, such as coherent legal frameworks and policy credibility, social objectives such as affordable energy or desirability of certain technologies, as well as environmental and economic objectives need to be considered together, to determine whether emission and removal targets shall be separate, how the targets should be set and how much flexibility for adjustment is permitted.

Future work should also explore more heterogeneous and diverse policy assumptions that better reflect the ambition levels of real world actors and the implications of separate targets for regional net-zero $CO_2$ or GHG targets. For that, the respective scope of emissions, local constraints to deployment as well as implications of burden sharing and fairness principles[49,50] are key.

Lastly, for designing optimal policy instruments, it is necessary to keep the long-term goal in mind[16] and not stop considerations at net-zero. In the case of CDR governance, after net-zero, net-negative emissions will likely be needed to return to the safe operating space after a temporal temerature overshoot[29]. In the literature, several proposals on how to operationalise net-negative emissions have been brought forward[23,51,52]. If targets on emissions and removals shall be separated, further analysis is needed on how it could help or hinder achieving net-negative emissions in the second half of this century.

In summary, if targets were to be separate, there are strong arguments to set emission reduction targets rather strict than too loose to avoid overemphasizing CDR and underinvestment into low-carbon technologies. This is in line with Amstrong and McLaren (2022)[53], who also call for very ambitious reduction targets with only moderate CDR contribution to achieve what they call 'a narrow convergence' to net-zero. Yet nCDR targets should ensure that climate relevant amounts are available in the second half of the century to enable net-negative emissions. The price on emissions, or alternative, equally stringent climate policy instruments, in 2030 are essential to bring the world on track to stay within the 1.5° carbon budget. It also reduces the overshoot and with that the necessary finance volumes to incentivize large amounts of net-negative emissions over the 2nd half of the 21st century. Hence, it is of utmost importance to not underestimate it in near-term policies.

## Methods
### Modelling framework
The model description is taken from (Strefler et al.)[54]. 'We use the global multi-regional energy-economy-climate model REMIND[27] Version [3.2.0][28] for our analysis. REMIND is open source and available on GitHub at https://github.com/remindmodel/remind. The technical documentation of the equation structure can be found at https://rse.pik-potsdam.de/doc/remind/3.2.0. In REMIND, each single region is modelled as a hybrid energy-economy system and is able to interact with the other regions by means of trade. Tradable goods are the exhaustible primary energy carriers coal, oil, gas and uranium, a composite good, and emission permits. The economy sector is modelled by a Ramsey-type growth model which maximizes utility, a function of consumption. Labour, capital, and end-use energy generate the macroeconomic output, i.e. GDP. The produced GDP covers the costs of the energy system, the macroeconomic investments, the export of a composite good, and consumption. The energy sector is described with high technological detail. It uses exhaustible and renewable primary energy carriers and converts them to final energies as electricity, heat, and fuels. Various conversion technologies are available, including technologies with carbon capture and storage (CCS). Regional annual CCS deployment is limited to 0.5% of total storage capacity. This limits total global CCS use to ~20 Gt $CO_2$ per yr.'

### Separate targets on residual emissions and CDR
In this study we set separate quantity targets on $CO_2$ emissions and nCDR (for explicit definition refer to Table 1) in 2050 such that global carbon neutrality is reached. For this we exclude negative emissions generated by nCDR technologies from the default tax on emissions and add a complementary removal price. The carbon price trajectories for both, emissions and removals, follow a Hotelling price path that increases at 5% per year and the starting value in 2025 is iteratively adapted such that the annual emission or removal target in 2050 is met. In the market efficient case the emission and removal prices are identical.

Land-use and land-use change $CO_2$ emissions as well as non-$CO_2$ GHGs are also penalised with the price on $CO_2$ emissions and are abated using exogenous marginal abatement cost curves derived from coupled REMIND-MAgPIE scenarios with comparably stringent climate protection. In all scenarios the carbon price on emission is sufficiently high to tap most of the abatement potential and hence the scenarios exhibit almost identical land-use change and non-$CO_2$ GHG contributions and we therefore forgo a detailed analysis of the respective emission reductions.

**Table 1 | Components of the separate emission targets divided into 'nCDR' and '$CO_2$ emissions' used throughout this study**

| $CO_2$ Removals (nCDR) | $CO_2$ Emissions (excluding nCDR) |
|---|---|
| BECCS (four supply side technology routes) | Gross energy |
| DACCS | Industrial processes |
| Enhanced Weathering | Land-use |
| Industry CDR (demand side CCS with carbon neutral fuels such as biofuels or synthetic fuels) | Land-use change (including positive emissions from deforestation or conversion of carbon rich land and removals from afforestation/reforestation) |

## CDR technology portfolio

The following nCDR options are available to REMIND v3.2.0: bioenergy with carbon capture and storage (BECCS) with 4 conversion routes (electricity, hydrogen, biogas and biodiesel), direct air carbon capture and storage (DACCS), enhanced weathering of rocks (EW) and industry CDR from combining carbon neutral fuels (i.e. bio- or synthetic fuels) with CCS in the industry sector. For techno-economic data on capture rates, costs, energy requirements and other relevant limitations of nCDR technologies see Supplementary Methods and Supplementary Tables 1-4. Total biomass availability is constrained in all scenarios to 100 EJ per yr for sustainability concerns[55], which is in all cases fully exploited after 2050. Therefore, the bioenergy impact on the land-system (and with that associated land-use change emissions) are almost identical across the scenario range. A sensitivity analysis with unconstrained biomass availability can be found in the Supplementary fig. 2.

Note that re-/afforestation cannot be treated the same way as the other CDR options. In the case of separate targets (and hence separate monetary incentives) of emission reduction and CDR it could lead to situations where clearing of existing forests in favour of reforestation is incentivised when the removal price is higher than the price on $CO_2$ emissions. We therefore exclude it from the total CDR target and focus solely on nCDR. We use the default REMIND-standalone setup with exogenous data on net-LULUCF emissions derived from coupled REMINDv3.2.0-MAgPIEv4.6.4 scenarios based on rcp2.0 and SSP2. For this study, net-LULUCF emissions are fully accounted for in the residual emission.

## Economic efficiency indicators

As the scenarios from the main text display variations in the respective cumulative emissions until 2050 (see Fig. 1b), pathways are not directly comparable with respect to their economic efficiency, as consumption losses result from the deviation from the model-internal economic efficient contribution of CDR to net-zero (i.e. from equal pricing) but even stronger from increasing mitigation effort with resulting lower cumulative emissions. As we explicitly want to assess the economic losses induced by the separation of targets, we have to subtract the consumption losses from increasing mitigation efforts. For that we use counterfactual scenarios that reach the same cumulative emissions but with only a single carbon market (i.e. equal prices) for emissions and removals. We calculate the difference of cumulative, discounted consumption losses from 2020–2050 (w.r.t. 2020 and at discount rate 5%) from the main scenarios with their corresponding counterfactual scenario. The additional consumption loss can then be attributed to the separation of targets and is displayed in (Fig. 3).

## Reporting summary

Further information on research design is available in the Nature Portfolio Reporting Summary linked to this article.

## Data availability

The specific model runs and scenario data as well as plotting routines for this study are archived at Zenodo under a CC-BY-4.0 license upon publication and is available under https://doi.org/10.5281/zenodo. 15367999[56]. Source data are provided with this paper.

## Code availability

The REMIND code is available under the GNU Affero General Public License, version 3 (AGPLv3) via GitHub https://github.com/ remindmodel/remind. We use a model version based on REMINDv3.2.0[28] that additionally includes a separate carbon market for novel CDR. The source code is archived together with the data at Zenodo[56] and is available on Github at https://github.com/amerfort/ remind/tree/SepMark_REMIND3.2.0.The technical documentation of the equation structure can be found at https://rse.pik-potsdam.de/ doc/remind/3.2.0.

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

## Acknowledgements

A.M has received funding from the European Union's Horizon Europe research and innovation programme under grant agreement No. 101081521 'Bridging current knowledge gaps to enable the UPTAKE of carbon dioxide removal methods' (UPTAKE project). G.A. has reveived funding from European Union's Horizon Europe research and innovation program under grant No. 101003536 'Earth system models for the future' (ESM2025 project). L.M. has received funding from European Union's Horizon Europe research and innovation program under grant no. 101056939 (RESCUE) and No. 101081193 (OptimESM). T.D. has received funding from the German Federal Ministry of Research, Technology and Space (BMBF) through the research project CDRSynTra (01LS2101A/C/G). The authors gratefully acknowledge the European Regional Development Fund (ERDF), the German Federal Ministry of Education and Research and the Land Brandenburg for supporting this project by providing resources on the high performance computer system at the Potsdam Institute for Climate Impact Research.

## Author contributions

A.M., J.S., G.A., N.B., T.D., E.K., G.L., L.M. and O.E., conceived the study. A.M., J.S., N.B., E.K., G.L. and O.E. conceived the experiments. A.M., J.S., G.A., N.B., T.D., E.K., G.L. and L.M. contributed to developing the energy-economy-climate model. A.M. performed the experiment. A.M., G.A. and L.M. performed data analysis and created the figures. A.M. wrote the manuscript with input and feedback from all authors.

## Funding

## Competing interests
The authors declare no competing interests.
