## [Transparent Peer Review file · Nature Communications]

Separating CO₂ emission from removal targets comes with limited cost impacts

Corresponding Author: Ms Anne Merfort

Version 0:

Reviewer comments:

Reviewer #1

(Remarks to the Author)

This is an important and very timely contribution, showing that a separation of CO₂ emission reduction targets from CO₂ removal targets for a 1.5 mitigation pathway implies only moderately higher costs than directly integrating CDR into a carbon market with a uniform carbon price. While the research is novel and innovative in that the cost implications of such a target separation have not yet been systematically investigated to date, the results are particularly relevant in light of the ongoing debate in the EU on whether to adopt a net- or a separate target for 2040, as well as parallel considerations of integrating CDR into the EU Emission Trading System (EU ETS). The paper also provides novel insights into the fiscal implications of different levels of CDR contribution to net-zero (and corresponding levels of residual emissions), highlighting in particular a heavy fiscal burden associated with high CDR contribution scenarios. Another striking finding is the fact that biomass use is almost identical across a wide range of low- to medium CDR contribution scenarios, suggesting that a low CDR target alone does not mitigate sustainability risks (however, as discussed below, we believe this finding requires further explication).

Given these important findings, we are generally supportive of the publication. However, we believe that some important issues need to be addressed first. Given the scope of our expertise in social science, we are necessarily constrained in our assessment of how key results were derived from the IAM as well as of the key assumptions underlying the model. Therefore, our comments mostly relate to the general analytical approach, how the results are situated within the literature in our field, and the extent to which the main conclusions and claims are supported by the results.

I. 52-53: As outlined here, the claim that a uniform carbon price in an integrated market is the most economically efficient solution to climate change mitigation critically hinges on the assumption that CO₂ removal and reduction are equivalent. However, as the literature you reference (McLaren et al. 2019) and others (e.g., Carton et al. 2021, <https://doi.org/10.3389/fclim.2021.664130>) highlight, this assumption is problematic in several ways, which should be acknowledged and made explicit by the authors.

I. 55-59: Target separation can enhance the political credibility of net-zero targets (as noted in the manuscript), but importantly, it can also prevent mitigation deterrence by sending political signals that deep emission reductions cannot be substituted by future nCDR deployment. We believe that this key rationale for target separation, i.e., managing expectations regarding substitution and fungibility of emission reductions and negative emissions, should be made more explicit from the outset, as it is currently missing from the introduction and only briefly alluded to in the conclusion. As this paragraph of the manuscript currently reads, you seem to question that target separation may limit mitigation deterrence because policy makers may overemphasize CDR, but please note that scholars calling for target separation have also advocated a “narrow convergence” toward net-zero (Armstrong and McLaren 2022, <https://doi.org/10.1017/S0892679422000521>), which we think the authors should recognize here.

I. 66-67: The reasons why an integrated market may provoke strategic behavior and lobbying may remain inaccessible to readers not familiar with this specific discussion. We suggest expanding and explaining this either here or at a later point in the manuscript, especially since you also highlight potential lobbying from high-emitting actors in the low CDR scenarios (see also our related, more general comment at the end of the review).

I. 75-84: This proposal for differentiated carbon prices is interesting, but its relevance to the analysis remains unclear to us, especially since you state in lines 85 and 86 that the discussion of carbon price differentiation in tax or emission trading schemes is not the focus of the paper. In addition, the exact workings of this proposal are not fully accessible without consulting the referenced literature, which is still forthcoming. In particular, the reader may wonder how compliance can be guaranteed by collateral if price developments and deployment capacities for nCDR are yet unknown, and to what extent a carbon central bank can act as a lender of last resort when future deployment capacities for nCDR may be significantly constrained. Also importantly, if the authors decide to retain this paragraph on differentiated carbon pricing within an emission trading scheme, we believe it is imperative to explain how this proposal relates to the existing literature on managing carbon removal in the EU ETS through a carbon central bank (i.e., Rickels et al. 2022, <https://doi.org/10.1016/j.erss.2022.102858>). Please also note that you often refer to the benchmark scenario with a uniform carbon price simply as the integrated market scenario, which may confuse readers as you argue here that an integrated market is compatible with differentiated prices and separate targets.

I. 87: What is meant by “systemic consequences”? This is a very generic term and should be replaced with a more precise alternative, especially since the paper focuses mainly on cost impacts.

I. 113: Should read “non-CO2 GHG emissions”.

I. 120-121: We question the assertion that stakeholders' concern that a subsidy for afforestation in excess of the carbon price creates perverse incentives was the only reason why they advocated for three separate targets in the EU consultation. Presumably more important is the intention to make a clear distinction between permanent removal and temporary sequestration. Unless you can provide evidence to support this claim, please consider revising this sentence.

I. 153-154: This is a surprising finding, given the concern that the scalability of DACCS is severely constrained by competition for renewable energy. Please provide more information on why this is not the case according to your model.

I. 197-213: This is also a striking finding that should be further explained in terms of the underlying assumptions that were made, particularly with respect to fuel consumption across sectors. In particular, to what extent have demand-side mitigation potentials as prominently discussed in the recent IPCC AR, been considered here (and if not, please justify why you have excluded these potentials).?

I. 200-203: You provide some interesting information here on different levels of residual emissions from the use of fossil fuels. However, other than that, we missed further details on the composition of residual emissions in different scenarios. This omission is particularly surprising given the growing interest in residual emissions, as evidenced by the recent literature on the subject that you cite.

Figure 3 a): This is a concise and helpful presentation of key findings, but due to the way you frame the near-term risks, it suggests that the risks associated with higher and lower CDR contribution scenarios are more or less symmetric. However, options for reducing emissions, including demand-side potentials, already exist, while nCDR methods are still at low technological readiness levels and not yet proven at scale, as highlighted in the paper. Please consider revising the figure to better reflect this asymmetric risk structure of the scenarios assessed.

I. 307-309: Please consider that such iterative adjustments may have certain benefits, but undermine the political signaling effects essential to limit mitigation deterrence, at least if adjustments are made frequently.

Finally, a more general issue that we think should be addressed throughout the manuscript, and especially in the conclusion: The paper finds that target separation has limited cost implications when compared to a uniform carbon price scenario which can be considered the most economically efficient option under a couple of assumptions. However, although not intended by the authors, the way that the results are presented may lead readers (and especially policy makers) to conclude that an integrated market with a uniform carbon price is still the way to go despite sustainability and mitigation deterrence risks, because it saves costs, while target separation options are more costly but similarly risky. Therefore, the risks and uncertainties associated with the benchmark scenario should be more explicitly communicated when presenting and discussing the results, especially in Figure 3 a) and in the conclusion. In particular, we believe it is important to emphasize that non-separation of targets may create expectations of high levels of CDR deployment by high-emitting actors, which – if frustrated – may lead to intense political pressure to adjust carbon price trajectories, thus undermining mitigation targets.

(Remarks on code availability)

Reviewer #2

(Remarks to the Author)

“I co-reviewed this manuscript with one of the reviewers who provided the listed reports. This is part of the Nature Communications initiative to facilitate training in peer review and to provide appropriate recognition for Early Career Researchers who co-review manuscripts.”

(Remarks on code availability)

Reviewer #4

(Remarks to the Author)

This paper aims to contribute to the important debate on if CDR policy should be governed with a separate target or be left to an integrated market with emissions reductions. The authors focus on the impacts, specifically economic efficiency, between the two approaches and use an IAM to calculate those costs and explain them. They argue that, on balance, the increased costs associated with separating targets or limiting CDR may be low enough to justify the other benefits of this policy. I think the approach is fine and paper well-presented, but it would be helpful if the authors could clarify the articles novelty.

The central argument of the paper seems to be that separate targets create credibility for supporting the deployment of CDR and can prevent the overuse of CDR. The authors argue that integrated market mechanisms may fail to fully account for the sustainability risks associated with novel CDR approaches, potentially leading to their overuse. However, there is no evidence to suggest that these mechanisms cannot be designed to internalise such risks, nor is there evidence that this issue is currently present in any compliance mechanisms globally. Perhaps lines 75 to 87 are an attempt to clarify that integrated markets can be valuable in certain conditions. It didn't really seem to come back into the paper though.

The key findings highlighted in the abstract are consistent with the effects of removing any technology from a least-cost optimisation model: reliance on that technology decreases, government expenditure on it reduces, and consequently, the overall solution becomes more expensive. This follows through to Figure 1, carbon prices will be higher if you restrict the use of a technology (e.g. DACCS) which could cap them. Also, the result that if we use more CDR then it will be more expensive isn't surprising, as we use up the cheapest options (BECCS) only the more expensive options remain (DACCS). If the authors aim to emphasise this point, a sensitivity analysis on the development of the CDR portfolio over time would be valuable. For instance, the 8 GtCO₂ tipping point could be highly sensitive to assumptions about technology learning curves, energy prices, deployment rates, and feedstock availability. From my reading, the authors reference studies that align closely with their own findings/relationships, which is helpful but might limit the novelty. I'm curious—what do these results really challenge? While these findings are undoubtedly valuable for policy, they don't seem especially groundbreaking from an academic standpoint.

On the paragraph lines 89 to 95 I'm not quite following the stated contribution. Arguably, Jenkins et al. (2021) in Joule is an earlier study that conceptualised the impact on mitigation costs of a separate target for novel CDR through their CTBO concept in an IAM. They aren't looking to limit CDR in that paper of course. The term 'appropriate residual emissions' seems a bit off to me. You have simply investigated a range of scenarios for limiting CDR use that could help policy makers understand what a desired level of residual emissions might be. This paper could be useful to support target setting itself but not the appropriate level of residual emissions.

It would be helpful to understand more about the justification for the scenarios that exceed the economically optimal 7GtCO₂ a year nCDR. This is quite a wide range and most of the discussion and focus seems to be at these fringe scenarios.

It would be useful for the authors to clarify if there was any methodological contribution from the study. My interpretation is that it's a published model using a few scenarios that limit CDR. Is that correct? It's just useful to know if there is a larger contribution here.

While the paper measures the impacts of separate targets being more expensive (less economically efficient). I'm not sure any of the evidence in the paper supports the conclusions that separate targets may be better at adding credibility to a quantity of CDR being available at a certain point in time or that it would reduce mitigation deterrence. In many cases targets on certain technologies deployment rates are revised to reflect the reality the effectiveness of the policy option. Also, policy credibility isn't necessarily measurable. So this cost/benefit trade-off mention on lines 225 to 227 isn't something we know for certain.

Minor Comments:

Page 1 line 49/50 it could be useful to cite some of the reviews of policy/commercialisation options to incentivise CDR at this point and perhaps provide some examples to give the reader a better sense of what you mean (e.g. fiscal incentives, market-based, public procurement).

On line 54 in the introduction. It could be made clearer why a separation of targets creates separate prices. If a CDR credit is inherently riskier than an emission reduction credit. Why would they trade at the same price in an integrated market? I understand there is one ETS price but why would a company buy CDR over an emissions reduction at the same price.

For line 65, it's debateable whether having a target for CDR increases planning security. In some ways it depends on the cost of non-compliance and the robustness of the political institutions that support the target. For example, targets to phase out ICE vehicles seem to always be changing.

On line 89 there is a full stop missing.

It could be helpful to have greater clarity why the CDR subsidy are narrower than the CO₂ price at line 146-148.

I found lines 227 and 228 hard to interpret. The whole paragraph could be much clearer.

On line 277 the word 'are' is missing.

The paragraphs can be very long. Particularly in the conclusion.
The referencing formatting is inconsistent. For some references the journal is included others it is not.

(Remarks on code availability)

Reviewer #5

(Remarks to the Author)

This was an interesting read! The manuscript presents very valuable and intriguing analysis, although the key argument and main results could be more clearly articulated (especially in the introduction and in the conclusions part). The core message would benefit from being more explicitly stated. The work is significant to the field and aligns well with established literature. While the manuscript supports its conclusions, some sections would benefit from additional evidence, as detailed in my comments. I did not identify any flaws in the data analysis, interpretation, or conclusions. However, since my expertise is limited to the field of law, I was unable to fully assess the methodology or its reproducibility.

Detailed comments:

Since my expertise lies in law and legal systems, I will concentrate my review on the legal and regulatory issues discussed in the manuscript.

Legal frameworks are crucial for investors when making decisions about projects with significant risk, such as many CDR projects. Furthermore, if CDR is not comprehensively regulated, it could potentially hinder emissions reductions (moral hazard or mitigation deterrence). However, the regulatory framework for CDR is still largely undeveloped. There are no directly applicable international legal frameworks, and national or regional frameworks are still in the early stages of development. Although the manuscript highlights that national strategies and policies are lagging behind the anticipated CDR deployment needed for 1.5°C scenarios, it would be beneficial to further consider the connection and relevance of between well-developed regulatory frameworks and the advancement of CDR technologies when discussing their role in achieving net-zero goals.

The introduction mentions that "a variety of policy instruments could be used to incentivize both ambitious emission reductions and CDR deployment. A uniform carbon price across all sectors and emissions is, in the absence of other externalities, the most economically efficient solution." It would be helpful if the authors could explain why a uniform carbon price is considered the most economically efficient option among the various policy instruments available.

The authors point to the scholarship that discusses proposed separate targets as a strategy to prevent mitigation deterrence. However, in addition to scientific and technological validation, and the development of appropriate policies and regulations, separate targets for emissions and removals would also need to be accepted, and developed, by policymakers and regulators, which raises questions about their legal feasibility and regulatory potential. Although the manuscript does not primarily build its argument around legal issues, it would still be beneficial to reference or point to the legal discussions related to CDR technologies. This is important because the targets themselves are established by law, and implementing separate targets would necessitate legal structuring. So far, also, the idea of separate targets has received limited attention from legal scholars, and there appears to be a scarcity of research on this topic. After a brief search, I identified a few texts that might be of interest to the authors. Please refer to the list below:

N. Ghaleigh & Justin Macinante, *Déjà vu All Over Again: Carbon Dioxide Removals (CDR) and Legal Liability*, *Journal of Environmental Law* 2023.

C. Kaupa, *Scrutinizing net zero: The legal problems of counting greenhouse gas emissions, removals and offsets together*, *RECIEL* 2022

M. Honegger, W. Burns & D.R. Morrow, *Is carbon dioxide removal 'mitigation of climate change'?* *RECIEL* 2021

E.A. Parson & H.J. Buck, *Large-Scale Carbon Dioxide Removal: The Problem of Phasedown*, *Global Environmental Politics* 2020.

Minor suggestions:

A brief explanation of what is meant by the sustainability risks associated with CDR would also be useful.

Also, it would be helpful if the authors could briefly explain what integrating CDR into an Emissions Trading Scheme (ETS) entails. This should include how and where this integration is discussed, as well as its relevance and potential benefits.

Could you clarify what is meant by "increasing climate target stringency" (line 146)? Does this refer to increased regulatory or policy strictness?

Additionally, proper references to the EU Net-Zero Industry Act are missing. It would be useful to include a reference that explains the Act's ambitious nature (lines 169 to 173). Also, please note that the Act addresses only a limited range of net-zero technologies and is not broadly applicable to all technologies.

The statement, "The deployment of specific technologies and their relative contribution can depend strongly on the CDR target" (line 190), could be expanded to acknowledge that the future deployment of CDR technologies on a feasible scale also depends on other significant factors. For example, the availability of critical raw materials, which are essential for the development of many CDR technologies, plays a crucial role.

"If net-zero is achieved using separate targets, ..." (line 216) This argument appears somewhat simplistic and would need to be supported by thorough reasoning about the potential and feasibility of separate targets. The same concern applies to the statement, "The separation of emission reduction and CDR targets could strengthen trust in political commitment, stabilize expectations, and set a clear signal for the pace and depth of the phase-out of fossil fuels" (line 250) and to the statement "In summary, there are strong arguments to set emission reduction targets rather strict than too loose to avoid overemphasizing CDR and underinvestment into low-carbon technologies." (line 307).

Could you clarify, briefly, what is meant by "from environmental side effects" (line 254)?

--

Based on these comments, I invite the authors to revise their manuscript to address specific concerns before a final decision is reached.

(Remarks on code availability)

Version 1:

Reviewer comments:

Reviewer #1

(Remarks to the Author)

The authors have satisfactorily addressed our comments in the revised and substantially improved manuscript. Within the scope of our expertise, we believe it is ready for release, and we look forward to seeing it published.

However, some minor points should still be addressed.

Line: 66-69: We believe that the mechanism through which mitigation deterrence work are not entirely adequately reflected in this formulation. Mitigation deterrence is not simply about shifting mitigation to the future, but expectations of future mitigation through nCDR delaying or obstructing near-term mitigation efforts. This could be addressed, for instance, with the following rephrasing: "It could provoke mitigation deterrence, when expectations of future mitigation through nCDR justify reduced near term emission reduction efforts, which could ultimately lead ...". The reference to Brad & Schneider 2023 cited in the discussion could be mentioned already at this point as well (along refs. 10-13).

We also suggest to refer to concerns over mitigation deterrence in the abstract, given its importance as a rationale for target separation and prominence in the introduction and discussion of the paper (as it stands, you only mention concerns over inadequate internalization of sustainability risks).

Line 82: consider replacing "in the future" with "in the long-term"

Line 83: consider replacing "On the other hand" with "At the same time" or similar (since you are not introducing an opposing but rather an additional argument for target separation)

Line 122: "same amount of nCDR" should be replaced by "corresponding amount of nCDR" to avoid confusion.

Figure 3a: still uses the old wording (CO2 price, CDR subsidy) and should be amended.

(Remarks on code availability)

Reviewer #2

(Remarks to the Author)

(Remarks on code availability)

Reviewer #4

(Remarks to the Author)

Thank you for addressing my questions and feedback.

I have some minor comments:

On lines 151, 152, 196 and 276 should this be termed 'removal price' for consistency. E.g 'First we discuss the variations of emission trajectories and corresponding carbon prices and nCDR subsidies'

Figure 1 panels c/d have not been updated to the new terminology of 'removal price' instead of subsidy. Figure 4 panel a says the same thing.

I find the explanation of the Biomass PE in figure 2 confusing. Why is it lower at 7Gt than at 2 or 4 Gt.

(Remarks on code availability)

Reviewer #5

(Remarks to the Author)

I have carefully read the revised version of the manuscript and appreciate that my comments were found useful. I can see that the contributors have thoughtfully considered my feedback, and I am satisfied with the revisions.

I have only one additional, minor comment: Although the manuscript does not focus on legal or regulatory issues per se, I would like to note, particularly in relation to the sentence "Additionally, scholars have started to outline how CDR could be included into already existing legal frameworks, such as the EU-ETS, as separate regulations are still largely missing" (line 309-), that the EU CFRP Regulation (Regulation (EU) 2024/3012) has entered into force since my previous review. This regulation establishes a voluntary certification framework for permanent carbon removals, carbon farming, and carbon storage in products within the EU. It sets quality criteria, verification rules, certification scheme requirements, and certified unit issuance rules. The regulation supports EU climate targets under the Paris Agreement, ensuring that certified removals contribute to EU objectives rather than third-party NDCs. It does not cover emissions under the EU ETS, except for CO₂ capture and storage from biofuels, bioliquids, and biomass fuels meeting sustainability criteria. Overall, governance efforts have not kept pace with the growing significance of CDR in climate policy, and regulatory frameworks remain underdeveloped, though they are evolving rapidly. While the CFRP Regulation has its uncertainties, it represents a step forward in this regulatory domain. I leave it to the authors to decide whether the new regulation warrants a mention in the paper.

(Remarks on code availability)

Revision: SEPARATING CO₂ EMISSION FROM REMOVAL TARGETS COMES WITH LIMITED COST IMPACTS

Summary of main changes:

- We changed the wording from "CO₂ price" and "CDR subsidy" to "price on emissions" and "price on removals" to distinguish the CDR subsidy in the context of my paper from the fiscal incentive mechanism of a general "subsidy", which could lead to confusion.
- We restructured the introduction and the argumentation from the existing literature on why targets should be separate. In short: i) Separate targets to prevent mitigation deterrence, ensure CDR development and enhance policy credibility, ii) separate targets to hedge against sustainability and environmental risks iii) separate targets to avoid high financing burdens on tax payers if taxing away high CDR rents seems impossible.
- We emphasise the aim of the study: We do not assess with the REMIND analysis *whether* targets should be separate, but we analyse the consequences, *if* targets were to be set separately by a regulator.
- We discuss the risks of separate targets (in the discussion) and the option for flexible adjustment along the first two objectives in the introduction. If the main concern is policy credibility, then targets should be binding and not renegotiable. If pricing in environmental side-effects and sustainability risks are the main concern, then targets could be iteratively adjusted for economic efficiency and intertemporal flexibility.
- The Supplemental Information now also contains a brief discussion on residual emissions across sectors and the sectoral mitigation options.
- We added additional explanations when reviewers requested more clarity and connected the work to the social science and legal and regulatory literature.

REVIEWER COMMENTS

Reviewer #1 (Remarks to the Author):

This is an important and very timely contribution, showing that a separation of CO₂ emission reduction targets from CO₂ removal targets for a 1.5 mitigation pathway implies only moderately higher costs than directly integrating CDR into a carbon market with a uniform carbon price. While the research is novel and innovative in that the cost implications of such a target separation have not yet been systematically investigated to date, the results are particularly relevant in light of the ongoing debate in the EU on whether to adopt a net- or a separate target for 2040, as well as parallel considerations of integrating CDR into the EU Emission Trading System (EU ETS). The paper also provides novel insights into the fiscal implications of different levels of CDR contribution to net-zero (and corresponding levels of residual emissions), highlighting in particular a heavy fiscal burden associated with high CDR contribution scenarios. Another striking finding is the fact that biomass use is almost identical across a wide range of low- to medium CDR contribution scenarios, suggesting that a low CDR target alone does not mitigate sustainability risks (however, as discussed below, we believe this finding requires further explication).

We sincerely thank the reviewers for their supportive feedback and the thoughtful, helpful and precise comments provided below.

Given these important findings, we are generally supportive of the publication. However, we believe that some important issues need to be addressed first. Given the scope of our expertise in social science, we are necessarily constrained in our assessment of how key results were derived from the IAM as well as of the key assumptions underlying the model. Therefore, our comments mostly relate to the general analytical approach, how the results are situated within the literature in our field, and the extent to which the main conclusions and claims are supported by the results.

Your perspective and the preciseness of your comments really helped us restructure both, the introduction and the discussion to strengthen important points and avoid misunderstandings.

l. 52-53: As outlined here, the claim that a uniform carbon price in an integrated market is the most economically efficient solution to climate change mitigation critically hinges on the assumption that CO₂ removal and reduction are equivalent. However, as the literature you reference (McLaren et al. 2019) and others (e.g., Carton et al. 2021, <https://doi.org/10.3389/fclim.2021.664130>) highlight, this assumption is problematic in several ways, which should be acknowledged and made explicit by the authors.

Excellent point. Your perspective helped us restructure the introduction. We use the reference you provide to introduce the discussion on separate targets with this fundamental problem of equivalence, which really helped the structure of the argumentation. We added the paragraph below that opens the argumentation both for and against separation of emission and removal targets.

Lines 61-69:

This, however, only holds true if CO₂ removals and emission reductions are regarded equivalent in their role for net-zero, which is problematic for a variety of reasons, especially when removals are to be delivered by the land-use sector⁷. CDR methods are associated with environmental side effects⁸ that pose risks for sustainability^{8,9}, such as potential leakage, toxicity of chemical sorbents or mineralisation methods or land-impacts on biodiversity, water availability or food security from biomass demand. Intertemporal equivalence, i.e., allowing continued emissions now to be offset by future removals, is seen especially problematic⁷. It could provoke mitigation deterrence, i.e., shifting mitigation to the future, which could ultimately lead to underachievement of climate targets if CDR methods fail to deliver. By challenging the equivalence of emission reductions and removals, multiple scholars advocate for a separation of targets^{7,10-13}.

l. 55-59: Target separation can enhance the political credibility of net-zero targets (as noted in the manuscript), but importantly, it can also prevent mitigation deterrence by sending political signals that deep emission reductions cannot be substituted by future nCDR deployment.

Very valid point. We added, that it is problematic to equate current emissions with future removals in the introduction:

l. 65-68 Intertemporal equivalence, i.e., allowing continued emissions now to be offset by future removals, is seen especially problematic⁷. It could provoke mitigation deterrence, i.e., shifting mitigation to the future, which could ultimately lead to underachievement of climate targets if CDR methods fail to deliver.

We believe that this key rationale for target separation, i.e., managing expectations regarding substitution and fungibility of emission reductions and negative emissions, should be made more explicit from the outset, as it is currently missing from the introduction and only briefly alluded to in the conclusion.

Thank you very much for pointing this out. With your valuable feedback we strengthened that point in the restructured paragraph of the introduction.

l. 79-88:

Furthermore, an integrated market leaves expectations of future CDR acceptability, desirability and ultimately availability unclear. This lack of clarity could reduce planning security for fossil emitters risking a lack of security for zero-carbon investments. This could provoke strategic behaviour and lobbying; in the near-term if fossil fuel emitters oppose the necessary speed of the transition and also in the future if expectations about large amounts of future removal capacities were frustrated and carbon prices jump sharply. On the other hand, the lack of clarity reduces planning security for CDR investors, yet ambitious CDR commitments are needed now to enable net-negative emissions in the second half of the century. Therefore, separate targets are proposed to manage expectations as a means to prevent mitigation deterrence¹⁴, ensure sufficient CDR development to achieve net-negative emissions later and enhance political credibility of net-zero targets^{11,12}, which might be more important for effective climate policy than economic efficiency¹⁵.

As this paragraph of the manuscript currently reads, you seem to question that target separation may limit mitigation deterrence because policy makers may overemphasize CDR, but please note that scholars calling for target separation have also advocated a “narrow convergence” toward net-zero (Armstrong and McLaren 2022, <https://doi.org/10.1017/S0892679422000521>), which we think the authors should recognize here.

Great point! This also connects to the addition to the introduction cited above (line 61-69). Due to your very helpful comments we restructured the introduction, giving more attention to the reasoning for separate targets from the existing literature.

We furthermore acknowledged the calling for a “narrow convergence” to net-zero from Armstrong and McLaren in the final paragraph of the discussion:

l. 373-376: In summary, if targets were to be separate, there are strong arguments to set emission reduction targets rather strict than too loose to avoid overemphasizing CDR and underinvestment into low-carbon technologies. This is in line with Armstrong and McLaren (2022)⁴⁹, who also call for very ambitious reduction targets with only moderate CDR contribution to achieve what they call “a narrow convergence” to net-zero.

l. 66-67: The reasons why an integrated market may provoke strategic behavior and lobbying may remain inaccessible to readers not familiar with this specific discussion. We suggest expanding and explaining this either here or at a later point in the manuscript, especially since you also highlight potential lobbying from high-emitting actors in the low CDR scenarios (see also our related, more general comment at the end of the review).

Thanks a lot for this valid point. We added a short explanation after the statement to make the argument more accessible.

l. 81-83: This could provoke strategic behaviour and lobbying; in the near-term if fossil fuel emitters oppose the necessary speed of the transition and also in the future if expectations about large amounts of future removal capacities were frustrated and carbon prices jump sharply.

l. 75-84: This proposal for differentiated carbon prices is interesting, but its relevance to the analysis remains unclear to us, especially since you state in lines 85 and 86 that the discussion of carbon price differentiation in tax or emission trading schemes is not the focus of the paper. In addition, the exact workings of this proposal are not fully accessible without consulting the referenced literature, which is still forthcoming. In particular, the reader may wonder how compliance can be guaranteed by collateral if price developments and deployment capacities for nCDR are yet unknown, and to what extent a carbon central bank can act as a lender of last resort when future deployment capacities for nCDR may be significantly constrained. Also importantly, if the authors decide to retain this paragraph on differentiated carbon pricing within an emission trading scheme, we believe it is imperative to explain how this proposal relates to the existing literature on managing carbon removal in the EU ETS through a carbon central bank (i.e., Rickels et al. 2022, <https://doi.org/10.1016/j.erss.2022.102858>).

Thanks a lot for pointing out that the paragraph is not sufficiently self-explaining. We decided to shorten it to not disrupt from the main message of this manuscript:

l. 98-102: It should be noted that while a separation of targets may lead to price differentiation, in reverse a price differentiation between emissions and removals does not exclude an integrated market approach. For example, there is emerging literature on how CDR could be included into the European Emissions Trading Scheme: Rickels et al. 2022²⁰ propose removal certificate reserves to ensure price stabilisation while Lessmann et al. 2024²¹ propose a new kind of certificate for future removals of current emissions.

Please also note that you often refer to the benchmark scenario with a uniform carbon price simply as the integrated market scenario, which may confuse readers as you argue here that an integrated market is compatible with differentiated prices and separate targets.

Excellent point! We carefully revised our wording to be consistent and more explicit not equating the benchmark scenario with an integrated market scenario. We added this explanation to the benchmark scenario:

l. 133-137: The scenario with 7 GtCO₂/yr as quantity targets for residual emissions and nCDR is the scenario from which identical prices in both markets emerge; i.e. the regulator perfectly guessed the market efficient contribution of CDR to net-zero. Note that in our model this is identical to a scenario with an integrated market and a uniform carbon price due to the model features, namely perfect foresight, certainty of future costs and the social planner. This scenario serves as a benchmark throughout the manuscript.

l. 87: What is meant by “systemic consequences”? This is a very generic term and should be replaced with a more precise alternative, especially since the paper focuses mainly on cost impacts.

Thanks for this hint. We changed the wording to be more specific here from “systemic consequences” to “consequences [...] on mitigation pathways” (line 108-109):

l. 113: Should read “non-CO₂ GHG emissions”.

Nice catch! Thanks a lot, we corrected it (line 138).

I. 120-121: We question the assertion that stakeholders' concern that a subsidy for afforestation in excess of the carbon price creates perverse incentives was the only reason why they advocated for three separate targets in the EU consultation. Presumably more important is the intention to make a clear distinction between permanent removal and temporary sequestration. Unless you can provide evidence to support this claim, please consider revising this sentence.

Thanks a lot for pointing this out. We fully agree that our wording in the original version was misleading and it was indeed not our intention to claim that stakeholders' concerns were about the perverse incentives in case of price differentiations. We also decided to focus in this section on the scenario design and moved the paragraph on the EU consultation and the suggestion of three separate targets to the introduction of the discussion section.

I 293-295: The idea of separate targets gained traction in the EU policy debate: The majority of stakeholders responding to the Public Consultation on the EU Climate Target for 2040³² advocated for separate targets on GHG emission reduction, industrial removals (i.e. nCDR) and nature based removals.

I. 153-154: This is a surprising finding, given the concern that the scalability of DACCS is severely constrained by competition for renewable energy. Please provide more information on why this is not the case according to your model.

Thank you very much for this interesting question. We added an explanation to the statement:

I. 180-184: On the other end of the spectrum, DACCS is an expensive but scalable option, and higher demands do not increase prices as much, due to two reasons. DACCS deployment per unit becomes cheaper with larger quantities due to learning and, by scenario design, a higher DACCS demand is paired with a less stringent target on emission reduction, relaxing competition for renewable energy, a key resource for large-scale DACCS deployment.

I. 197-213: This is also a striking finding that should be further explained in terms of the underlying assumptions that were made, particularly with respect to fuel consumption across sectors. In particular, to what extent have demand-side mitigation potentials as prominently discussed in the recent IPCC AR, been considered here (and if not, please justify why you have excluded these potentials).?

Thanks a lot for pointing this out. We included a brief summary of demand-sector mitigation strategies displayed in the scenarios in the Supplemental Information. However, we did not include additional demand reductions that would result from significant shifts in societal consumption patterns, such as dietary changes. This is due to the fact that we want to keep the scenarios as comparable as possible to each other. I.e., we want to keep the underlying drivers identical such that we are able to extract the effect of the separation of markets. Other studies have focused on how to further reduce residual emissions and hence CDR reliance with demand-side mitigation potentials and this dimension of the problem is not the focus of this study. We added this explanation to the main text:

I: 240-244: Note that CDR reliance can be reduced significantly by demand-side mitigation options that essentially reduce total energy or food consumption²³. Yet, this study focuses on the effect of market separation and we compare CDR contributions across scenarios with otherwise identical drivers. Hence, we do not include additional

demand reductions other than what arises endogenously from the model as a cheaper option compared to alternative decarbonisation strategies (see SI).

I. 200-203: You provide some interesting information here on different levels of residual emissions from the use of fossil fuels. However, other than that, we missed further details on the composition of residual emissions in different scenarios. This omission is particularly surprising given the growing interest in residual emissions, as evidenced by the recent literature on the subject that you cite.

Great suggestion. In the main text we want to emphasise the consequences of separate targets and focus the analysis on nCDR. Yet, in the Supplemental Information we added a section on the sectoral mitigation strategies and respective sectoral residual emissions. We added a reference to the SI in the main text:

I: 228-229 Further information on the level of residual emissions and especially their distribution across sectors can be found in the Supplemental Information (S-Figure 1).

Additional information in SI:

Supplementary-Figure 1: Emissions are drastically reduced in all sectors and all scenarios, with the biggest share consistently remaining in transport due to the high remaining demand in liquid fuels for aviation and shipping. In all scenarios, the power sector is almost fully decarbonised and the demand sectors' transformations heavily rely on electrification to a similar degree across scenarios, suggesting that the potential is almost fully tapped in all scenarios. To bring down residual emissions, fossil liquids are increasingly replaced by a mixture of biofuels and synthetic fuels, with higher shares of synthetic fuels in scenarios with higher emission reduction (i.e. lower allowed residual emissions). Note that in REMINDv3.2.0, the distribution of different types of liquids across sectors is arbitrary and no conclusions can be drawn from a certain type of liquid being used in one sector rather than in another. The industry sector furthermore relies on hydrogen with similar shares on final energy across scenarios. To a certain extent, the energy system also relies on demand reductions in those sectors that rely on liquid fuel and feedstocks (transport and industry). This is cheaper than an even faster or earlier scale-up of synfuels.

Figure 3 a): This is a concise and helpful presentation of key findings, but due to the way you frame the near-term risks, it suggests that the risks associated with higher and lower CDR contribution scenarios are more or less symmetric. [However, options for reducing emissions, including demand-side potentials, already exist, while nCDR methods are still at low technological readiness levels and not yet proven at scale, as highlighted in the paper.*

Addressed below] Please consider revising the figure to better reflect this asymmetric risk structure of the scenarios assessed.

This is a great point, thanks for sparking this interesting debate. We agree with you that the risks are not symmetric and made this now explicit in the figure caption. But we don't see how the figure suggests that high and low CDR contributions are equally risky and as we still want to keep all the arguments, we decided to not revise the figure, but add explicitly to the figure caption:

Figure caption: We find generally greater risks for high nCDR contributions. Note however, that these risks are not easily comparable and strongly depend on an individual risk perception.

We also want to point out that we already advocate for low-CDR targets due to the entailed benefits in the discussion section:

l. 373-381 In summary, if targets were to be separate, there are strong arguments to set emission reduction targets rather strict than too loose to avoid overemphasizing CDR and underinvestment into low-carbon technologies. This is in line with Armstrong and McLaren (2022)⁵⁰, who also call for very ambitious reduction targets with only moderate CDR contribution to achieve what they call “a narrow convergence” to net-zero. Yet nCDR targets should ensure that climate relevant amounts are available in the second half of the century to enable net-negative emissions. The price on emissions, or alternative, equally stringent climate policy instruments, in 2030 are essential to bring the world on track to stay within the 1.5° carbon budget. It also reduces the overshoot and with that the necessary finance volumes to incentivize large amounts of net-negative emissions over the 2nd half of the 21st century. Hence, it is of utmost importance to not underestimate it in near-term policies.

[...] However, options for reducing emissions, including demand-side potentials, already exist, while nCDR methods are still at low technological readiness levels and not yet proven at scale, as highlighted in the paper. [...]

The fundamental question is: what is perceived to be riskier? This crucially depends on the respective perspective. Relying on technologies that are yet unproven at large scale or relying on global, deep transformations in societal consumption patterns. While demand reductions may be desirable due to numerous positive side-effects, it remains yet unclear how they can be realised at the necessary scale and pace. We believe that there is so far no scientific consensus on what strategy is less risky.

As these questions about various risk-perceptions and objectives are still open, we added a paragraph to the discussion that a broader negotiation between different perspectives is urgently needed:

l. 358-362: We call for the interdisciplinary scientific community to evaluate and negotiate between different risk perspectives. The political objectives, such as coherent legal frameworks and policy credibility, social objectives such as affordable energy or desirability of certain technologies, as well as environmental and economic objectives need to be considered together, to determine whether emission and removal targets shall be separate, how the targets should be set and how much flexibility for adjustment is permitted.

l. 307-309: Please consider that such iterative adjustments may have certain benefits, but undermine the political signaling effects essential to limit mitigation deterrence, at least if adjustments are made frequently.

Thank you very much for this important point, which lead us to restructure the argumentation. We now highlight in the introduction that there are different objectives, why

targets should be separate (lines 74-94):

In short: 1) avoid mitigation deterrence and enhance policy credibility 2) accounting for sustainability risks and environmental side effects 3) fiscal considerations and potentially high burdens on tax payers.

And in the discussion, we pick this point up again and conclude that depending on the objective, iterative adjustments may or may not be desirable, as you rightfully pointed out:

I. 291-294: If the main concern is the avoidance of mitigation deterrence and the enhancement of policy credibility, then targets have to be binding and non-negotiable. On the other hand, if environmental side-effects are the main concern, a mechanism for iterative adjustments of the two targets, once more knowledge on side effects and future costs becomes available, might be an option to increase intertemporal flexibility.

Finally, a more general issue that we think should be addressed throughout the manuscript, and especially in the conclusion: The paper finds that target separation has limited cost implications when compared to a uniform carbon price scenario which can be considered the most economically efficient option under a couple of assumptions. However, although not intended by the authors, the way that the results are presented may lead readers (and especially policy makers) to conclude that an integrated market with a uniform carbon price is still the way to go despite sustainability and mitigation deterrence risks, because it saves costs, while target separation options are more costly but similarly risky. Therefore, the risks and uncertainties associated with the benchmark scenario should be more explicitly communicated when presenting and discussing the results, especially in Figure 3 a) and in the conclusion.

Thank you very much for this key observation! Due to your valuable help we clarified in several parts of the paper that the main analysis with REMIND does not compare the case of an integrated market and the case of separate markets with separate targets. But rather we address the key argument against separate targets, which is economic efficiency. We quantify efficiency losses induced by this separation and since they are limited, we conclude that this key argument might be less important than previously assumed.

We clarified the aim of the study here:

I:99-103: Instead, while an IAM by itself is not able to answer the question whether targets for emissions and removals should be separated, we investigate the trade-off with economic efficiency if targets were separate.

We furthermore analyse the consequences on mitigation pathways of deviations from the market efficient contribution of CDR to net-zero.

And we added this explicitly to the conclusion:

I 300-305: The REMIND modelling framework is not equipped to capture many of the externalities and market imperfections raised above; i.e. myopic investments, strategic actors and uncertainty about future costs and potentials of CDR methods. Therefore, we do not aim to answer the question whether targets should be separate but instead we assess the consequences of setting targets and with that potentially diverging from the market efficient contribution of CDR to net-zero. But as we find that efficiency losses due to the separation of carbon markets are moderate, this prominent argument for integrated carbon markets may be overestimated.

In particular, we believe it is important to emphasize that non-separation of targets may create expectations of high levels of CDR deployment by high-emitting actors, which – if

frustrated – may lead to intense political pressure to adjust carbon price trajectories, thus undermining mitigation targets.

Thanks again for this valuable feedback. Due to your thoughtful, insightful and helpful comments we believe that with the restructured introduction and several additional explanations we now cover this point as mentioned above in the introduction (l. 77-83) and in the discussion:

l. 282-289: There are two main lines of reasoning in the literature for separating targets for emissions and removals: Firstly, many scholars argue, a separation could prevent mitigation deterrence^{10,11,15,32}. Because it could strengthen trust in political commitments^{10,11}, enable evaluations of ambitions in mitigation plans³³ and stabilise expectations. Separate targets could set a clear signal for the pace and depth of the phase-out of fossil fuels^{11,12} and increase planning security for CDR suppliers to ensure sufficient CDR development such that net-negative emissions can be achieved after net-zero. Secondly, a separation of emissions and removals is necessary, if damages from environmental side effects^{8,34,35} are not reflected in the prices but are valued higher than the loss of economic efficiency from deviating from the economic optimum.

Reviewer #2 (Remarks to the Author):

“I co-reviewed this manuscript with one of the reviewers who provided the listed reports. This is part of the Nature Communications initiative to facilitate training in peer review and to provide appropriate recognition for Early Career Researchers who co-review manuscripts.”

Thank you very much for contributing to this extremely helpful review. Your joint review inspired us to restructure the argumentation both in the introduction as well as in the discussion, which we believe is much better and clearer now. You also pointed out aspects where we were not clear enough, such as the “benchmark scenario” or the results of the EU-consultation on 2040 targets with the separate land-sector target. This was crucial to avoid misunderstandings.

Reviewer #4 (Remarks to the Author):

This paper aims to contribute to the important debate on if CDR policy should be governed with a separate target or be left to an integrated market with emissions reductions. The authors focus on the impacts, specifically economic efficiency, between the two approaches and use an IAM to calculate those costs and explain them. They argue that, on balance, the increased costs associated with separating targets or limiting CDR may be low enough to justify the other benefits of this policy. I think the approach is fine and paper well-presented, but it would be helpful if the authors could clarify the articles novelty.

Thank you very much for your rather positive evaluation of our manuscript. And thanks a lot for pointing out that the novelty of this work has to be made more explicit. We strengthened this at the end of the introduction and also added an explanation of the methodological advancement.

l 110-115: To the best of our knowledge, a quantitative analysis of 1.5°C mitigation pathways with separate targets and deliberately chosen levels for residual emissions and associated CDR has not

yet been conducted in the scientific literature. In this study, for the first time, we integrate this policy framework into an Integrated Assessment Model (IAM). By adding a separate carbon market for nCDR, we analyse the economic consequences of such a policy, which is not possible in a conventional set-up with an integrated carbon market and a uniform carbon price but constraints on CDR.

The central argument of the paper seems to be that separate targets create credibility for supporting the deployment of CDR and can prevent the overuse of CDR. The authors argue that integrated market mechanisms may fail to fully account for the sustainability risks associated with novel CDR approaches, potentially leading to their overuse. However, there is no evidence to suggest that these mechanisms cannot be designed to internalise such risks, nor is there evidence that this issue is currently present in any compliance mechanisms globally.

Thank you very much for pointing this out. We acknowledge the possibility of such internalisation of risks and externalities.

l 103-105: In this paper we do not discuss and comment on the emerging literature on how to deal with carbon price differentiation in tax or emission trading schemes. Neither do we discuss mechanisms to internalise the risks, externalities and imperfections in an integrated market.

Furthermore, we state the arguments **for** an integrated market more precisely in the discussion:

l 295-299: In contrast, an integrated market is advocated for due to economic efficiency and the high uncertainty about future quantities that makes target setting now difficult³⁷. Additionally, scholars have started to outline how CDR could be included into already existing legal frameworks, such as the EU-ETS^{22,37,38}, as separate regulations are still largely missing³⁷. Comprehensive and solid legal frameworks are a precondition for fast nCDR deployment, such as liability for storage integrity³⁹, MRV (monitoring, reporting, verification)⁴⁰ or carbon accounting of removals⁴¹.

Perhaps lines 75 to 87 are an attempt to clarify that integrated markets can be valuable in certain conditions. It didn't really seem to come back into the paper though.

As a proper connection of this proposal to the rest of the work would have required substantial additional explanations (see also comments from the other reviewers), we shortened this paragraph.

l. 98-102: It should be noted that while a separation of targets may lead to price differentiation, in reverse a price differentiation between emissions and removals does not exclude an integrated market approach. For example, there is emerging literature on how CDR could be included into the European Emissions Trading Scheme: Rickels et al. 2022²⁰ propose removal certificate reserves to ensure price stabilisation while Lessmann et al. 2024²¹ propose a new kind of certificate for future removals of current emissions.

The key findings highlighted in the abstract are consistent with the effects of removing any technology from a least-cost optimisation model: reliance on that technology decreases, government expenditure on it reduces, and consequently, the overall solution becomes more

expensive. This follows through to Figure 1, carbon prices will be higher if you restrict the use of a technology (e.g. DACCS) which could cap them. Also, the result that if we use more CDR then it will be more expensive isn't surprising, as we use up the cheapest options (BECCS) only the more expensive options remain (DACCS). If the authors aim to emphasise this point, a sensitivity analysis on the development of the CDR portfolio over time would be valuable. For instance, the 8 GtCO₂ tipping point could be highly sensitive to assumptions about technology learning curves, energy prices, deployment rates, and feedstock availability. From my reading, the authors reference studies that align closely with their own findings/relationships, which is helpful but might limit the novelty. I'm curious—what do these results really challenge? While these findings are undoubtedly valuable for policy, they don't seem especially groundbreaking from an academic standpoint.

We appreciate that you raise that point. We agree with your statement and modified the abstract to highlight the key findings more, i.e. the limited cost impact and the fact that some key risks of large-scale CDR deployment commonly raised in the literature cannot necessarily be reduced by a small CDR target. However, we also still want to highlight the other important implications of high and low CDR scenarios. We believe, even though they are (luckily) not challenging any well-established knowledge from the energy system's transformation community they are nevertheless crucial for the interdisciplinary audience of Nature Communications to fully understand the implications of separating targets.

l 27-33: We find that efficiency losses are moderate and the argument of economic efficiency for an integrated market may be overstated. If targets were to be separate, limiting CDR comes with increasing emission prices but also significant benefits: lower cumulative emissions, a lower financial burden for public finance and limited reliance on geologic CO₂ storage. Yet a small CDR target alone might not be sufficient to mitigate the risks associated with high biomass demand or nascent carbon capture technologies, as for deeper decarbonisation these might be needed for bio- and synfuels. Proposed targets should also ensure sufficient CDR deployment to achieve net-negative emissions in the second half of 21st century.

On the paragraph lines 89 to 95 I'm not quite following the stated contribution. Arguably, Jenkins et al. (2021) in Joule is an earlier study that conceptualised the impact on mitigation costs of a separate target for novel CDR through their CTBO concept in an IAM. They aren't looking to limit CDR in that paper of course.

Thank you for mentioning Jenkins et al. (2021), which we now also acknowledge in our introduction as related work on alternative CDR incentive. We would, however, like to argue that this study is conceptually very different as it did not choose a separate target for novel CDR, which you also point out in your comment. Instead, within the framework of CTBOs the level of CDR emerges as an outcome of the market for "Carbon Storage Units". The consequences of setting a target based on societal or political objectives are not and cannot be assessed. Furthermore, they do not use an Integrated Assessment Model for their analysis but just an emulator, derived from IAM scenario results.

The term 'appropriate residual emissions' seems a bit off to me. You have simply investigated a range of scenarios for limiting CDR use that could help policy makers understand what a desired level of residual emissions might be. This paper could be useful to support target setting itself but not the appropriate level of residual emissions.

Thanks for pointing that out. We agree and changed the wording to “deliberately chosen levels of residual emissions” (line 111)

It would be helpful to understand more about the justification for the scenarios that exceed the economically optimal 7GtCO₂ a year nCDR. This is quite a wide range and most of the discussion and focus seems to be at these fringe scenarios.

Thanks for raising this. We added a short justification in the main text

l 128-132: As the market efficient outcome is difficult to predict with current knowledge, policymakers may choose targets that diverge from what would emerge from an integrated market. Therefore, we span the scenario range from 2 to 12 GtCO₂/yr of residual emissions (corresponding to ~5% up to ~27% with respect to 2019 global CO₂ emissions²⁷, see SI for the underlying rationale) and the same amount of compensating nCDR in 2050.

and added the explanation of the scenario rationale to the SI.

Scenario rationale

As the future market efficient CDR contribution to net-zero is hard to predict with only current knowledge, there is a high chance that in case policymakers were to choose separate targets for emissions and removals now, they could over- or underestimate the CDR contribution. While in our modelling framework, 7 GtCDR/yr is the market efficient contribution, in reality this could very well be much lower. As we want to explore both, the consequences of higher and lower than market efficient contributions, we span the scenario range from 2-12 GtCO₂/yr to still cover plausible ranges of relative deviations. Note that while the 12 GtCO₂/yr scenario seems excessive in its reliance on CDR we still find it plausible that policymakers may overestimate optimal future CDR contributions by a factor of 2 (e.g. in a world, where the market efficient contribution of CDR to net-zero is really small such as 1 GtCO₂/yr).

It would be useful for the authors to clarify if there was any methodological contribution from the study. My interpretation is that it's a published model using a few scenarios that limit CDR. Is that correct? It's just useful to know if there is a larger contribution here.

Thanks again for pointing this out. As already mentioned in response to a previous comment, we clarified the methodological advancement at the end of the introduction:

l 112-115: In this study, for the first time, we integrate this policy framework into an Integrated Assessment Model (IAM). By adding a separate carbon market for novel CDR, we are able to directly analyse the economic consequences of such a policy, which is not possible in a conventional set-up with an integrated carbon market and a uniform carbon price but constraints on CDR.

While the paper measures the impacts of separate targets being more expensive (less economically efficient). I'm not sure any of the evidence in the paper supports the conclusions that separate targets may be better at adding credibility to a quantity of CDR being available at a certain point in time or that it would reduce mitigation deterrence.

We fully agree, thanks for your comment. The argument for political credibility is from the social science literature that we cite in the introduction and discussion section. We don't conclude from our analysis that separate targets add credibility, as the model is not able to assess “credibility”. Rather we investigate the (economic and energy system) consequences

of the proposal of separate targets. To better reflect that in the paper, we highlighted it more precisely in the introduction:

l. 105-107: Instead, while an IAM by itself is not able to answer the question *whether* targets for emissions and removals should be separated, we investigate the trade-off with economic efficiency *if* targets were separate.

As well as in the discussion in a more extended version:

l. 299-304: The REMIND modelling framework is not equipped to capture many of the externalities and market imperfections raised above; i.e. myopic investments, strategic actors and uncertainty about future costs and potentials of CDR methods. Therefore, we do not aim to answer the question *whether* targets should be separate. Instead we assess the consequences of setting targets and with that potentially diverging from the market efficient contribution of CDR to net-zero. We find that efficiency losses due to the separation of carbon markets are moderate, suggesting that this prominent argument for integrated carbon markets may be overestimated.

In many cases targets on certain technologies deployment rates are revised to reflect the reality the effectiveness of the policy option. Also, policy credibility isn't necessarily measurable. So this cost/benefit trade-off mention on lines 225 to 227 isn't something we know for certain.

This is a very important point, thank you very much. We restructured the introduction along two main lines of reasoning from the existing literature on separate targets: i) Separate targets to prevent mitigation deterrence and enhance policy credibility and ii) separate targets to hedge against sustainability and environmental risks (lines 74-88) and pick it up in the discussion (lines 282-289). We then conclude with the valuable point in the discussion section that you raised here.

l. 291-294: If the main concern is the avoidance of mitigation deterrence and the enhancement of policy credibility, then targets have to be binding and non-negotiable. On the other hand, if environmental side-effects are the main concern, a mechanism for iterative adjustments of the two targets, once more knowledge on side effects and future costs becomes available, might be an option to increase intertemporal flexibility.

Minor Comments:

Page 1 line 49/50 it could be useful to cite some of the reviews of policy/commercialisation options to incentivise CDR at this point and perhaps provide some examples to give the reader a better sense of what you mean (e.g. fiscal incentives, market-based, public procurement).

Very good suggestion. We included the categories from the literature overview from Hickey et al. to provide readers with an initial overview. Following that, we decided to rename what we in our original work called a "CDR subsidy" to "price for removal" to avoid any confusion with the general fiscal incentive scheme of a "subsidy".

On line 54 in the introduction. It could be made clearer why a separation of targets creates separate prices. If a CDR credit is inherently riskier than an emission reduction credit. Why would they trade at the same price in an integrated market? I understand there is one ETS price but why would a company buy CDR over an emissions reduction at the same price.

Thanks for this question. We added some explanation to the introduction:

l.56-60: From a purely economic perspective, a uniform carbon price in all sectors and on all emissions is, in absence of other externalities, the economically efficient solution⁶, which would be delivered by an integrated market for emissions and removals. For incentivising CDR this means: CO₂ emissions should be avoided to the point, until it is cheaper to compensate the next ton of CO₂ by CDR. I.e., the price paid for CDR should be equal to the carbon price on emissions, such that marginal abatement costs equal the marginal supply costs of CDR.

For line 65, it's debateable whether having a target for CDR increases planning security. In some ways it depends on the cost of non-compliance and the robustness of the political institutions that support the target. For example, targets to phase out ICE vehicles seem to always be changing.

Valid point, thanks a lot for this interesting comment. We restructured our argumentation as mentioned above and believe that this covers also this concern raised in here. I.e. we acknowledge in the discussion that for objective i), iterative adjustments are detrimental and undermine policy credibility, while for ii) they might be a desirable way to enhance economic efficiency and intertemporal flexibility and we point out that future works including all different perspectives should elaborate, *whether* targets should be separate.

On line 89 there is a full stop missing.

Thanks, nice catch.

It could be helpful to have greater clarity why the CDR subsidy are narrower than the CO₂ price at line 146-148.

Thank you very much for making us aware that this was not sufficiently clear yet. We rephrased this paragraph to be more precise:

l. 177-184: The main reason for the lower price sensitivity of CDR is that even for a low CDR demand scenario there are no low-cost nCDR options available. This is, because sustainable biomass is always limited, and the high demand for biofuels in low CDR demand scenarios forces the more expensive Fischer-Tropsch-BECCS technology into the CDR portfolio. On the other end of the spectrum, DACCS is an expensive but scalable option, and higher demands do not increase prices as much, due to two reasons. DACCS deployment per unit becomes cheaper with larger quantities due to learning and, by scenario design, a higher DACCS demand is paired with a less stringent target on emission reduction, relaxing competition for renewable energy, a key resource for large-scale DACCS deployment.

I found lines 227 and 228 hard to interpret. The whole paragraph could be much clearer.

Thanks for pointing that out. We restructured the paragraph and added additional information to make our statement about additional consumption losses more assessable.

l. 259- 272: To quantify the efficiency losses we calculate consumption losses. But as scenarios differ in their

cumulative emissions (**Error! Reference source not found.**), we cannot directly compare the consumption losses across the scenario range, because lower cumulative emissions generally result in higher consumption losses regardless of whether emission and removal markets are integrated or separate. To isolate the consumption losses introduced by the separation of targets (and the deviation from equal prices) from the consumption losses caused by achieving lower cumulative emissions (**Error! Reference source not found.**), the additional consumption loss is calculated with respect to counterfactual scenarios that are not shown here but described in the Methods. These scenarios achieve the same respective cumulative CO₂ budget until 2050 but with a uniform carbon price on emissions and removals (see Methods). We find that for moderate deviations (4-10 GtCO₂/yr) from the equal-pricing contribution of nCDR to net-zero we observe only moderate efficiency losses of <10% additional consumption loss (**Error! Reference source not found.**). In absolute terms, it is an increase from 2.6% to 3.1% of consumption loss compared to a scenario with only current policies for the 12 GtCO₂/yr (from 2020-2050 with respect to continued current policies) and only 4.3% to 4.5% in 2 GtCO₂/yr scenario. This is in line with Strefler et al.³² who also found only moderately increasing mitigation costs for moderate limitations on CDR in a uniform carbon pricing framework.

On line 277 the word 'are' is missing.

Thanks, nice catch. added it.

The paragraphs can be very long. Particularly in the conclusion.

True, thanks for pointing it out. We reduced the lengths of paragraphs throughout the manuscript.

The referencing formatting is inconsistent. For some references the journal is included others it is not.

Thanks for pointing this out. We corrected that, now all references show the respective journal.

Reviewer #5 (Remarks to the Author):

This was an interesting read! The manuscript presents very valuable and intriguing analysis, although the key argument and main results could be more clearly articulated (especially in the introduction and in the conclusions part). The core message would benefit from being more explicitly stated. The work is significant to the field and aligns well with established literature. While the manuscript supports its conclusions, some sections would benefit from additional evidence, as detailed in my comments. I did not identify any flaws in the data analysis, interpretation, or conclusions. However, since my expertise is limited to the field of law, I was unable to fully assess the methodology or its reproducibility.

We are very grateful for your general appreciation and your very helpful comments. We restructured our argumentation and made the aim and the main conclusions of our paper more explicit (see comments below).

Detailed comments:

Since my expertise lies in law and legal systems, I will concentrate my review on the legal

and regulatory issues discussed in the manuscript.

This is a very valuable addition and we appreciate your unique perspective!

Legal frameworks are crucial for investors when making decisions about projects with significant risk, such as many CDR projects. Furthermore, if CDR is not comprehensively regulated, it could potentially hinder emissions reductions (moral hazard or mitigation deterrence). However, the regulatory framework for CDR is still largely undeveloped. There are no directly applicable international legal frameworks, and national or regional frameworks are still in the early stages of development. Although the manuscript highlights that national strategies and policies are lagging behind the anticipated CDR deployment needed for 1.5°C scenarios, it would be beneficial to further consider the connection and relevance of between well-developed regulatory frameworks and the advancement of CDR technologies when discussing their role in achieving net-zero goals.

Excellent point. We added the reference to the lack of legal frameworks in several parts of the manuscript. In the introduction:

l 46-48: While CDR methods play a significant role in climate change mitigation pathways, as of today both industrial scale-up as well as international and national legal frameworks and policies lag behind the envisioned CDR deployment in 1.5°C scenarios⁴.

In the section on nCDR deployment, when we discuss scale-up:

l 196-199: CDR subsidies in the 10 and 12 GtCO₂/yr scenarios in 2040 are already high enough (> 200\$/tCO₂) to incentivise significant contributions from EW. This is due to the fact that EW deployment relies on infrastructure for mining, grinding and transportation of material that already exists today and therefore EW could be scaled up in shorter time periods, yet regulatory and legal frameworks are largely missing.

In the discussion:

l 308-312: In contrast, an integrated market is advocated for due to economic efficiency and the high uncertainty about future quantities that makes target setting now difficult³⁷. Additionally, scholars have started to outline how CDR could be included into already existing legal frameworks, such as the EU-ETS^{22,37,38}, as separate regulations are still largely missing³⁷. Comprehensive and solid legal frameworks are a precondition for fast nCDR deployment, such as liability for storage integrity³⁹, MRV (monitoring, reporting, verification)⁴⁰ or carbon accounting of removals⁴¹.

The introduction mentions that "a variety of policy instruments could be used to incentivize both ambitious emission reductions and CDR deployment. A uniform carbon price across all sectors and emissions is, in the absence of other externalities, the most economically efficient solution." It would be helpful if the authors could explain why a uniform carbon price is considered the most economically efficient option among the various policy instruments available.

Thanks for this question. We added some explanation to the introduction:

l.56-60: From a purely economic perspective, a uniform carbon price in all sectors and on all emissions is, in absence of other externalities, the economically efficient solution⁶, which would be delivered by an integrated market for emissions and removals. For incentivising CDR this means: CO₂ emissions should be avoided to the

point, until it is cheaper to compensate the next ton of CO₂ by CDR. I.e., the price paid for CDR should be equal to the carbon price on emissions, such that marginal abatement costs equal the marginal supply costs of CDR.

The authors point to the scholarship that discusses proposed separate targets as a strategy to prevent mitigation deterrence. However, in addition to scientific and technological validation, and the development of appropriate policies and regulations, separate targets for emissions and removals would also need to be accepted, and developed, by policymakers and regulators, which raises questions about their legal feasibility and regulatory potential. Although the manuscript does not primarily build its argument around legal issues, it would still be beneficial to reference or point to the legal discussions related to CDR technologies. This is important because the targets themselves are established by law, and implementing separate targets would necessitate legal structuring. So far, also, the idea of separate targets has received limited attention from legal scholars, and there appears to be a scarcity of research on this topic. After a brief search, I identified a few texts that might be of interest to the authors. Please refer to the list below:

N. Ghaleigh & Justin Macinante, *Déjà vu All Over Again: Carbon Dioxide Removals (CDR) and Legal Liability*, *Journal of Environmental Law* 2023.

Thanks a lot for pointing us to this part of the CDR literature. We included the suggested manuscript and highlighted the importance of legal frameworks for CDR scale-up again in the discussion together with additional literature, applicable to the case we make in this analysis:

1 290-294: In contrast, an integrated market is advocated for due to economic efficiency and the high uncertainty about future quantities that makes target setting now difficult³⁷. Additionally, scholars have started to outline how CDR could be included into already existing legal frameworks, such as the EU-ETS^{22,37,38}, as separate regulations are still largely missing³⁷. Comprehensive and solid legal frameworks are a precondition for fast nCDR deployment, **such as liability for storage integrity**³⁹, MRV (monitoring, reporting, verification)⁴⁰ or carbon accounting of removals⁴¹.

Additional literature on CDR regulatory frameworks:

MRV: Lebling, K., Riedl, D. & Leslie-Bole, H. *Measurement, Reporting, and Verification for Novel Carbon Dioxide Removal in US Federal Policy*. WRIPUB (2024) doi:10.46830/wriwp.23.00044.

Carbon accounting of removals: Nordahl, S. L. et al. *Carbon accounting for carbon dioxide removal*. *One Earth* 7, 1494–1500 (2024).

C. Kaupa, *Scrutinizing net zero: The legal problems of counting greenhouse gas emissions, removals and offsets together*, *RECIEL* 2022

Thanks a lot for sharing this paper. However, it is primarily concerned with impermanent, land-based removals and the questionable offsets from additional emission reductions elsewhere. We exclude the former from our analysis and the latter is not applicable for the solely global perspective we adopted for this manuscript.

M. Honegger, W. Burns & D.R. Morrow, *Is carbon dioxide removal ‘mitigation of climate change’?* *RECIEL* 2021

While this is a very valuable manuscript, we already presume, like many other researchers and in particular everyone in the field of Integrated Assessment Modelling, that CDR is part of climate change mitigation.

E.A. Parson & H.J. Buck, Large-Scale Carbon Dioxide Removal: The Problem of Phasedown, Global Environmental Politics 2020.

We thank you for this thoughtful suggestion. This manuscript is concerned with a phase-down of a CDR sector, after net-negative emission targets have been reached. However, in this analysis we only focus on net-zero and we do not discuss the implications for net-negative emissions. An indefinite continuation of net-zero CO₂ emissions is necessary to stabilise global mean temperatures and hence from the scenarios we derive, a phase-down of CDR is not advisable.

Minor suggestions:

A brief explanation of what is meant by the sustainability risks associated with CDR would also be useful.

Thanks a lot for this request. We mean the environmental side-effects that are associated with large scale CDR deployment that pose sustainability risks. We also added examples on commonly discussed environmental side effects to the respective paragraph in the introduction for better clarity:

l. 63-65: CDR methods are associated with environmental side effects⁸ that pose risks for sustainability^{8,9}, such as potential leakage, toxicity of chemical sorbents or mineralisation methods or land-impacts on biodiversity, water availability or food security from biomass demand.

Also, it would be helpful if the authors could briefly explain what integrating CDR into an Emissions Trading Scheme (ETS) entails. This should include how and where this integration is discussed, as well as its relevance and potential benefits.

Thanks for the interesting request. Because a proper connection of the proposal, where the integration to the ETS with a new type of certificates was mentioned, to the rest of the work would have required substantial additional explanations (see also comments from the other reviewers), we decided to shorten this paragraph to avoid confusion.

l. 98-102: It should be noted that while a separation of targets may lead to price differentiation, in reverse a price differentiation between emissions and removals does not exclude an integrated market approach. For example, there is emerging literature on how CDR could be included into the European Emissions Trading Scheme: Rickels et al. 2022²¹ propose removal certificate reserves to ensure price stabilisation while Lessmann et al. 2024²² propose a new kind of certificate for future removals of current emissions.

Could you clarify what is meant by "increasing climate target stringency" (line 146)? Does this refer to increased regulatory or policy strictness?

Thanks for spotting this subject-specific formulation. For greater clarity we rephrased this to "increasing mitigation effort" (line 172).

Additionally, proper references to the EU Net-Zero Industry Act are missing.

Nice catch, thanks for pointing this out. We added the proper reference to the EU Net-Zero Industry Act (line 190).

It would be useful to include a reference that explains the Act's ambitious nature (lines 169 to 173).

Thanks for pointing out that some clarity is missing on this claim. We rephrased it to highlight that not the Act itself but the 50 Mt CCS target is ambitious. For this we add information on currently installed capacities and the implied growth rates to reach the target.

l. 192-195: This CCS target is already ambitious, especially compared to the total capacity of projects within the EU that are operating or under construction as of 2023 of only 2.65 MtCO₂/yr³⁰ which requires annual growth rates of 52%.

Also, please note that the Act addresses only a limited range of net-zero technologies and is not broadly applicable to all technologies.

You are completely right and we clarified already in the original manuscript that the Act does not cover all technologies and acknowledge that it's envisioned CCS capacities may not be available for CDR.

The statement, "The deployment of specific technologies and their relative contribution can depend strongly on the CDR target" (line 190), could be expanded to acknowledge that the future deployment of CDR technologies on a feasible scale also depends on other significant factors. For example, the availability of critical raw materials, which are essential for the development of many CDR technologies, plays a crucial role.

Thanks for this very interesting point. We were only referring to what we observe from the model scenarios in this section and within the scenario rationale, the deployment of specific technologies and their relative contribution does depend on the CDR target. We found your comment very helpful and acknowledge at the end of the section that there are further constraints that are not captured by the model:

l. 219-221: Furthermore, there are constraints to deployment that are not captured by the model such as the availability of critical raw material, public opposition or the lack of regulatory frameworks, that could severely constrain future CDR deployment.

"If net-zero is achieved using separate targets, ..." (line 216) This argument appears somewhat simplistic and would need to be supported by thorough reasoning about the potential and feasibility of separate targets.

Thanks for this very helpful comment! At this part of the paper we do not want to discuss, whether targets should be separate. We just assess the consequences, if they were. Due to your valuable feedback, we emphasise the aim of the study more explicitly and added explanation.

In the introduction:

l:101-103: Instead, while an IAM by itself is not able to answer the question whether targets for emissions and removals should be separated, we investigate the trade-off with economic efficiency if targets were separate.

And in the conclusion:

l 313-318: The REMIND modelling framework is not equipped to capture many of the externalities and market imperfections raised above; i.e. myopic investments, strategic actors and uncertainty about future costs and potentials of CDR methods. Therefore, we do not aim to answer the question whether targets should be separate but instead we assess the consequences of setting targets and with that potentially diverging from the market efficient contribution of CDR to net-zero. We find that efficiency losses due to the separation of carbon markets are moderate, suggesting that this prominent argument for integrated carbon markets may be overestimated.

The same concern applies to the statement, "The separation of emission reduction and CDR targets could strengthen trust in political commitment, stabilize expectations, and set a clear signal for the pace and depth of the phase-out of fossil fuels" (line 250) and to the statement "In summary, there are strong arguments to set emission reduction targets rather strict than too loose to avoid overemphasizing CDR and underinvestment into low-carbon technologies." (line 307).

We truly appreciate the point you raised. Additional to the point we addressed above, we furthermore added a disclaimer that separate targets may only strengthen trust in political commitment, if targets were binding and renegotiations are disallowed:

l. 304-307: If the main concern is the avoidance of mitigation deterrence and the enhancement of policy credibility, then targets have to be binding and non-negotiable. On the other hand, if environmental side-effects are the main concern, a mechanism for iterative adjustments of the two targets, once more knowledge on side effects and future costs becomes available, might be an option to increase intertemporal flexibility.

We furthermore added a paragraph to the discussion that the question of whether targets should be separate is still open:

l. 358-362: We call for the interdisciplinary scientific community to evaluate and negotiate between different risk perspectives. The political objectives, such as coherent legal frameworks and policy credibility, social objectives such as affordable energy or desirability of certain technologies, as well as environmental and economic objectives need to be considered together, to determine whether emission and removal targets shall be separate, how the targets should be set and how much flexibility for adjustment is permitted.

Could you clarify, briefly, what is meant by "from environmental side effects" (line 254)?

Thank you very much for pointing that out. We added examples of commonly discussed environmental side effects of large-scale CDR deployment to the description (as described in our reply to your first minor points comment).

GENERAL REMARKS FROM THE AUTHORS

Additionally to addressing the reviewer comments below we decided to adapt the wording: In this revision we now use the formulation “separate targets” consistently across the manuscript to avoid confusion. Previously we used “separate targets” and “separate markets” interchangeably, which is unambiguously true in the modelling framework used in this analysis but not with respect to real-world implementation.

REVIEWER COMMENTS

Reviewer #1 (Remarks to the Author):

The authors have satisfactorily addressed our comments in the revised and substantially improved manuscript. Within the scope of our expertise, we believe it is ready for release, and we look forward to seeing it published.

We thank the reviewer team for the effort and care they put into this review and we are looking forward to seeing it published soon too.

However, some minor points should still be addressed.

Line: 66-69: We believe that the mechanism through which mitigation deterrence work are not entirely adequately reflected in this formulation. Mitigation deterrence is not simply about shifting mitigation to the future, but expectations of future mitigation through nCDR delaying or obstructing near-term mitigation efforts. This could be addressed, for instance, with the following rephrasing: “It could provoke mitigation deterrence, when expectations of future mitigation through nCDR justify reduced near term emission reduction efforts, which could ultimately lead ...”. The reference to Brad & Schneider 2023 cited in the discussion could be mentioned already at this point as well (along refs. 10-13).

We also suggest to refer to concerns over mitigation deterrence in the abstract, given its importance as a rationale for target separation and prominence in the introduction and discussion of the paper (as it stands, you only mention concerns over inadequate internalization of sustainability risks).

We truly appreciate your help in adequately reflecting *mitigation deterrence*. We gratefully adopted your recommendation.

Line 82: consider replacing “in the future” with “in the long-term”

Thanks, we agree and changed the wording as suggested.

Line 83: consider replacing “On the other hand” with “At the same time” or similar (since you are not introducing an opposing but rather an additional argument for target separation)

You are right, we adopted your suggestion, thank you.

Line 122: “same amount of nCDR” should be replaced by “corresponding amount of nCDR” to avoid confusion.

Thanks, another helpful improvement of wording. Much appreciated.

Figure 3a: still uses the old wording (CO2 price, CDR subsidy) and should be amended.

Thank you very much. We now made sure that the new wording is consistently used across the manuscript, including Figure 1 and 3.

Reviewer #2 (Remarks to the Author):

Reviewer #4 (Remarks to the Author):

Thank you for addressing my questions and feedback.

I have some minor comments:

On lines 151, 152, 196 and 276 should this be termed ‘removal price’ for consistency. E.g ‘First we discuss the variations of emission trajectories and corresponding carbon prices and nCDR subsidies’

Thank you very much. We now made sure that the new wording is consistently used across the manuscript, including Figure 1, Figure 3 and lines L147, L152, L197.

Figure 1 panels c/d have not been updated to the new terminology of 'removal price' instead of subsidy. Figure 4 panel a says the same thing.

Thank you, please refer to our previous response above.

I find the explanation of the Biomass PE in figure 2 confusing. Why is it lower at 7Gt than at 2 or 4 Gt.

Thank you very much for raising this question. We added an additional explanation for clarification:

L223-226: Total biomass use is lowest in the equal-pricing net-zero formulation and increases stronger for high CDR targets (blue scenarios) due to higher BECCS demand. But also for low CDR targets (pink scenarios) (Figure 2 panel b) biomass demand increases due to increased pressure to decarbonise remaining liquid fuels that can no longer be offset by CDR.

Reviewer #5 (Remarks to the Author):

I have carefully read the revised version of the manuscript and appreciate that my comments were found useful. I can see that the contributors have thoughtfully considered my feedback, and I am satisfied with the revisions.

Thank you again for your helpful change of perspective and constructive feedback throughout the review process.

I have only one additional, minor comment: Although the manuscript does not focus on legal or regulatory issues per se, I would like to note, particularly in relation to the sentence "Additionally, scholars have started to outline how CDR could be included into already existing legal frameworks, such as the EU-ETS, as separate regulations are still largely missing" (line 309-), that the EU CFRP Regulation (Regulation (EU) 2024/3012) has entered into force since my previous review. This regulation establishes a voluntary certification framework for permanent carbon removals, carbon farming, and carbon storage in products within the EU. It sets quality criteria, verification rules, certification scheme requirements, and certified unit issuance rules. The regulation supports EU climate targets under the Paris Agreement, ensuring that certified removals contribute to EU objectives rather than third-party NDCs. It does not cover emissions under the EU ETS, except for CO₂ capture and storage from biofuels, bioliquids, and biomass fuels meeting sustainability criteria. Overall, governance efforts have not kept pace with the growing significance of CDR in climate policy, and regulatory frameworks remain underdeveloped, though they are evolving rapidly. While the CFRP Regulation has its uncertainties, it represents a step forward in this regulatory domain. I leave it to the authors to decide whether the new regulation warrants a mention in the paper.

Thank you so much for pointing this out. We decided to include a brief description of the CRCF to acknowledge this important step for CDR governance:

L318-L322: An important step forward is the EU Carbon Removals and Carbon Farming Certification (CRCF) Regulation (Regulation (EU) 2024/3012)⁴² that was published in November 2024. This regulation establishes a voluntary certification framework for permanent carbon removals, carbon farming, and carbon storage in products within the EU by setting quality criteria, verification rules, certification scheme requirements, and certified unit issuance rules.